# COUNTERNET: END-TO-END TRAINING OF PREDICTION AWARE COUNTERFACTUAL EXPLANATIONS

## ABSTRACT

Counterfactual (or CF) explanations are a type of local explanations for Machine Learning (ML) model predictions, which offer a contrastive case as an explanation by finding the smallest changes (in feature space) to the input data point, which will lead to a different prediction by the ML model. Existing CF explanation techniques suffer from two major limitations: (i) all of them are *post-hoc methods* designed for use with proprietary ML models — as a result, their procedure for generating CF explanations is uninformed by the training of the ML model, which leads to misalignment between model predictions and explanations; and (ii) most of them rely on solving separate time-intensive optimization problems to find CF explanations for each input data point (which negatively impacts their runtime). This work makes a novel departure from the prevalent post-hoc paradigm (of generating CF explanations) by presenting *CounterNet*, an *end-to-end* learning framework which integrates predictive model training and the generation of counterfactual (CF) explanations into a single pipeline. We adopt a block-wise coordinate descent procedure which helps in effectively training CounterNet's network. Our extensive experiments on multiple real-world datasets show that CounterNet generates high-quality predictions, and consistently achieves 100% CF validity and low proximity scores (thereby achieving a well-balanced cost-invalidity trade-off) for any new input instance, and runs 3X faster than existing state-of-the-art baselines.

## 1 INTRODUCTION

A counterfactual (CF) explanation offers a contrastive case — to explain the prediction made by a Machine Learning (ML) model on data point $x$, CF explanation methods find a new *counterfactual* example $x'$, which is similar to $x$ but gets a different (or opposite) prediction from the ML model. From an end-user perspective, CF explanation methods[1] (Wachter et al., 2017) may be more preferable (as compared to other methods of explaining ML models), as they can be used to offer recourse to vulnerable groups. For example, if a person applies for a loan and gets rejected by a bank's ML algorithm, CF explanation methods can suggest corrective measures to the loan applicant, which can be incorporated in a future loan application to improve their chances of getting an approved loan.

Generating high-quality CF explanations is a challenging problem because of the need to balance the *cost-invalidity trade-off* (Rawal et al., 2020) between: (i) the *invalidity*, i.e., the probability that a CF example is invalid, or it does not achieve the desired (or opposite) prediction from the ML model; and (ii) the *cost of change*, i.e., the $L_1$ norm distance between input instance $x$ and CF example $x'$. Figure 1 illustrates this trade-off by showing three different CF examples for an input instance $x$. If *invalidity* is ignored (and optimized only for *cost of change*), the generated CF example can be trivially set to $x$ itself. Conversely, if *cost of change* is ignored (and optimized only for *invalidity*), the generated CF example can be set to $x'_2$ (or any sufficiently distanced instance with different labels). More generally, CF examples with high (low) invalidities usually imply low (high) *cost of change*. To optimally balance this trade-off, it is critical for CF explanation methods to have access to the

---

[1]CF explanations are closely related to algorithmic recourse (Ustun et al., 2019) and contrastive explanations (Dhurandhar et al., 2018). Although these terms are proposed under different contexts, their differences from CF explanations have been blurred (Verma et al., 2020; Stepin et al., 2021), i.e. these terms are used interchangeably.

decision boundary of the ML model, without which finding a near-optimal CF explanation (i.e., $x_1'$) is difficult. For example, it is difficult to distinguish between $x_1'$ (a valid CF example) and $x_0'$ (an invalid CF example) without prior knowledge of the decision boundary.

Existing CF explanation methods suffer from three major limitations. First, to our best knowledge, all prior methods belong to the *post-hoc* explanation paradigm, i.e., they assume a trained black-box ML model as input. This post-hoc assumption has certain advantages, e.g., post-hoc explanation techniques are often agnostic to the particulars of the ML model, and hence, they are generalizable enough to interpret any *third-party* proprietary ML model. However, we argue that in many real-world scenarios, the model-agnostic approach provided by post-hoc CF explanation methods is not desirable. With the advent of data regulations that enshrine the "*Right to Explanation*" (e.g., EU-GDPR (Wachter et al., 2017)), service providers are required by law to communicate both the decision outcome (i.e., the ML model's prediction) and its actionable implications (i.e., a CF explanation for this prediction) to an end-user. In these scenarios, the post-hoc assumption is overly limiting, as service providers can build specialized CF explanation techniques that can leverage the knowledge of their particular ML model to generate higher-quality CF explanations. Second, in the post-hoc CF explanation

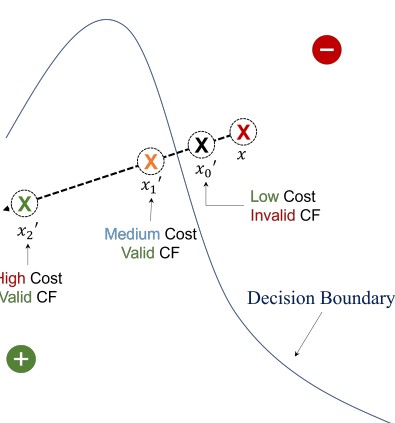

Figure 1: Illustration of the cost-invalidity trade-off in CF explanations for binary classification problems.

paradigm, the optimization procedure that finds CF explanations is completely uninformed by the ML model training procedure (and the resulting decision boundary). Consequently, such a post-hoc procedure does not properly balance the cost-invalidity trade-off (as explained above), causing shortcomings in the quality of the generated CF explanations (as shown in Section 4). Finally, most CF explanation methods are very slow — they search for CF examples by solving a separate time-intensive optimization problem for each input instance (Wachter et al., 2017; Mothilal et al., 2020; Karimi et al., 2021), which is not viable in time-constrained environment, as the runtime is a critical factor when deployed to end-user facing devices (Zhao et al., 2018; Arapakis et al., 2021).

**Contributions.** We make a novel departure from the prevalent post-hoc paradigm of generating CF explanations by proposing *CounterNet*, a learning framework that combines the training of the ML model and the generation of corresponding CF explanations into a single end-to-end pipeline (i.e., from input to prediction to explanation). CounterNet has three contributions:

- Unlike post-hoc approaches (where CF explanations are generated after the ML model is trained), CounterNet uses a (neural network) model-based CF generation method, enabling the joint training of its CF generator network and its predictor network. At a high level, CounterNet's CF generator network takes as input the learned representations from its predictor network, which is jointly trained along with the CF generator. This joint training is key to achieving a well-balanced cost-invalidity trade-off (as we show in Section 4).

- We theoretically analyze CounterNet's objective function to show two key challenges in training CounterNet: (i) poor convergence of learning; and (ii) a lack of robustness against adversarial examples. To remedy these issues, we propose a novel block-wise coordinate descent procedure.

- We conduct extensive experiments which show that CounterNet generates CF explanations with ~100% validity and low cost of change (~9.8% improvement to baselines), which shows that CounterNet balances the cost-invalidity trade-off significantly better than baseline approaches. In addition, this joint-training procedure does not sacrifice CounterNet's predictive accuracy and robustness. Finally, CounterNet runs orders of magnitude (~3X) faster than baselines.

## 2  RELATED WORK

Prior explanation techniques for ML models include LIME (Ribeiro et al., 2016), SHAP (Lundberg & Lee, 2017), saliency maps (Selvaraju et al., 2017; Sundararajan et al., 2017; Smilkov et al., 2017), which highlight attribution importance for each data instance. Further, *case-based methods* provide

(similar) data samples as model explanations (Guidotti et al., 2018; Molnar et al., 2020; Chen et al., 2019; Koh & Liang, 2017). Our work is most closely related to prior literature on *CF explanation* methods, which focuses on finding new instances that lead to different predicted outcomes (Wachter et al., 2017; Verma et al., 2020; Karimi et al., 2020; Stepin et al., 2021). CF explanations are preferred by human end-users as these explanations provide actionable recourse in many domains (Binns et al., 2018; Miller, 2019; Bhatt et al., 2020). Almost all prior work in this area belongs to the post-hoc CF explanation paradigm, which we categorize into *non-parametric* and *parametric* methods.

**Non-parametric methods.** Non-parametric methods aim to find a counterfactual explanation without the use of parameterized models. Wachter et al. (2017) proposed *VanillaCF* which generates CF explanations by minimizing the distance between the input instance and the CF example, while pushing the new prediction towards the desired class. Other algorithms, built on top of *VanillaCF*, optimize other aspects, such as recourse cost (Ustun et al., 2019), fairness (Von Kügelgen et al., 2022), diversity (Mothilal et al., 2020), closeness to the data manifold (Van Looveren & Klaise, 2019), causal constraints (Karimi et al., 2021), uncertainty (Schut et al., 2021), and robustness to model shift (Upadhyay et al., 2021). However, this line of work is inherently post-hoc and relies on solving a separate optimization problem for each input instance. Consequently, running them is time-consuming, and their post-hoc nature leads to poor balancing of the cost-invalidity trade-off.

**Parametric Methods.** These methods use parametric models (e.g., a neural network model) to generate CF explanations. For example, Pawelczyk et al. (2020); Joshi et al. (2019) generate CF explanations by perturbing the latent variable of a variational autoencoder (VAE) model. Yang et al. (2021); Singla et al. (2020); Nemirovsky et al. (2022) and Mahajan et al. (2019); Rodríguez et al. (2021); Guyomard et al. (2022) train generative models (GAN and VAE, respectively) to produce CF explanations for a trained ML model. However, these methods are still post-hoc in nature, and thus, they also suffer from poorly balanced cost-invalidity trade-offs. Contrastingly, we depart from this post-hoc paradigm, which leads to a greater alignment between CounterNet's predictions and CF explanations. Note that Ross et al. (2021) propose a recourse-friendly ML model by integrating recourse training during predictive model training. However, their work does not focus on generating CF explanations. In contrast, we focus on generating predictions and CF explanations simultaneously.

## 3 THE PROPOSED FRAMEWORK: COUNTERNET

Unlike prior work, our proposed framework CounterNet relies on a novel integrated architecture which combines predictive model training and counterfactual explanation generation into a single optimization framework. Through this integration, we can simultaneously optimize the accuracy of the trained predictive model and the quality of the generated counterfactual explanations. Formally, given an input instance $x \in \mathbb{R}^d$, CounterNet aims to generate two outputs: (i) the ML prediction component outputs a prediction $\hat{y}_x$ for input instance $x$; and (ii) the CF explanation generation component produces a CF example $x' \in \mathbb{R}^d$ as an explanation for input instance $x$. Ideally, the CF example $x'$ should get a different (and often more preferable) prediction $\hat{y}_{x'}$, as compared to the prediction $\hat{y}_x$ on the original input instance $x$ (i.e., $\hat{y}_{x'} \neq \hat{y}_x$). In particular, if the desired prediction output is binary-valued $(0, 1)$, then $\hat{y}_x$ and $\hat{y}_{x'}$ should take on opposite values (i.e., $\hat{y}_x + \hat{y}_{x'} = 1$).

### 3.1 NETWORK ARCHITECTURE

Figure 2 illustrates CounterNet's architecture which includes three components: (i) an encoder network $h(\cdot)$; (ii) a predictor network $f(\cdot)$; and (iii) a CF generator network $g(\cdot)$. During training, each input instance $x \in \mathbb{R}^d$ is first passed through the encoder network to generate a dense latent vector representation of $x$ (denoted by $z_x = h(x)$). Then, this latent representation is passed through both the predictor network and the CF generator network. The predictor network outputs a softmax representation of the prediction $\hat{y}_x = f(z_x)$. To generate CF examples, the CF generator network takes two pieces of information: (i) the final representation of the predictor network $p_x$ (before it is passed through the softmax layer), and (ii) the latent vector $z_x$ (which contains a dense representation of the input $x$). These two vectors are concatenated to produce the final latent vector $z'_x = p_x \oplus z_x$, which is passed through the CF generator network to produce a CF example $x' = g(z'_x)$. Note that passing the representation of predictor network $p_x$ through the CF generator network implicitly conveys information about the decision boundary to the CF generation procedure, who leverages this knowledge to find high-quality CF examples $x'$.

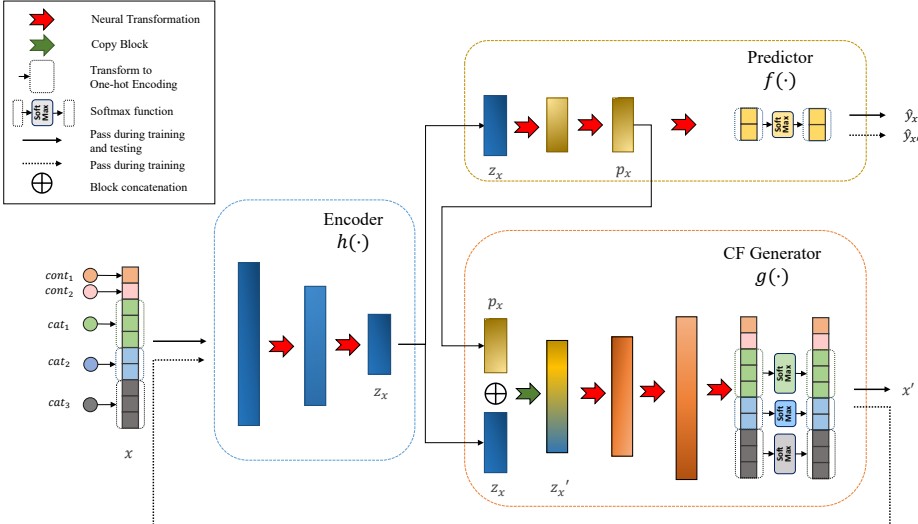

Figure 2: CounterNet contains three components: an encoder to transform the input into a dense latent vector, a predictor network to output the prediction, and a CF generator to produce explanations.

Furthermore, to ensure that the CF generator network outputs *valid* CF examples (i.e., $\hat{y}_x \neq \hat{y}_{x'}$), the output of the CF generator network $x'$ is also passed back as an input through the encoder and predictor networks when training CounterNet. This additional feedback loop (from the output of CF generator network back into the encoder and predictor networks) optimizes the *validity* of generated CF examples. As such, we can now train the entire network in a way such that the predictor network outputs opposite predictions $\hat{y}_x$ and $\hat{y}_{x'}$ for the input instance $x$ and the CF example $x'$, respectively. Note that this "*feedback loop*" connection is only needed during training, and is removed at test time. This design aims to achieve a better balance on the cost-invalidity tradeoff (as shown in Section 4).

**Design of Encoder, Predictor & CF Generator**. All three components in CounterNet's architecture consist of a multi-layer perception (MLP)[2]. The encoder network in CounterNet consists of two feed-forward layers that down-sample to generate a latent vector $z \in \mathbb{R}^k$ (s.t. $k < d$). The predictor network passes this latent vector $z$ through two feed-forward layers to produce the predictor representation $p$. Finally, the predictor network outputs the probability distribution over predictions with a fully-connected layer followed by a softmax layer. On the other hand, the CF generator network takes the final latent representation $z' = z \oplus p$ as an input, and up-samples (using two feed-forward layers) to produce CF examples $x' \in \mathbb{R}^d$. Each feed-forward neural network layer inside CounterNet uses LeakyRelu activation functions (Xu et al., 2015) followed by a dropout layer (Srivastava et al., 2014). Note that the number of feed-forward layers, the choice of activation function, etc., were hyperparameters that were optimized using grid search. See Appendix B.4 for implementation details.

**Handling Categorical Features**. To handle categorical features, we customize CounterNet's architecture for each dataset. First, we transform all categorical features in each dataset into numeric features via one-hot encoding. In addition, for each categorical feature, we add a softmax layer after the final output layer in the CF generator network (Figure 2), which ensures that the generated CF examples respect the one-hot encoding format (as the output of the softmax layer will sum up to 1). Finally, we normalize all continuous features to the $[0, 1]$ range before training.

## 3.2 COUNTERNET OBJECTIVE FUNCTION

CounterNet has three objectives: (i) *predictive accuracy* - the predictor network should output accurate predictions $\hat{y}_x$; (ii) *counterfactual validity* - CF examples $x'$ produced by the CF generator network should be valid, i.e., they get opposite predictions from the predictor network (e.g. $\hat{y}_x + \hat{y}_{x'} = 1$); and (iii) *minimizing cost of change* - minimal modifications should be required to change input instance $x$

---

[2]CounterNet can work with alternate neuronal blocks, e.g., convolution, attention, although effective training of these neuronal blocks demands additional efforts (see Appendix H).

to CF example $x'$. Thus, we formulate this multi-objective minimization problem to optimize the parameter of overall network $\theta$:

$$\underset{\theta}{\mathrm{argmin}} \frac{1}{N} \sum_{i=1}^{N} \left[ \lambda_1 \cdot \underbrace{(y_i - \hat{y}_{x_i})^2}_{\text{Prediction Loss } (\mathcal{L}_1)} + \lambda_2 \cdot \underbrace{\left(\hat{y}_{x_i} - \left(1 - \hat{y}_{x'_i}\right)\right)^2}_{\text{Validity Loss } (\mathcal{L}_2)} + \lambda_3 \cdot \underbrace{(x_i - x'_i)^2}_{\text{Change Loss } (\mathcal{L}_3)} \right] \quad (1)$$

where $N$ denotes the number of instances in our dataset, $(\lambda_1, \lambda_2, \lambda_3)$ are hyper-parameters to balance the three loss components, the prediction loss $\mathcal{L}_1$ denotes the mean squared error (MSE) between the actual and the predicted labels ($y_i$ and $\hat{y}_{x_i}$ on instance $x_i$, respectively), which aims to maximize predictive accuracy. Similarly, the validity loss $\mathcal{L}_2$ denotes the MSE between the prediction on instance $x_i$ (i.e., $\hat{y}_{x_i}$), and the opposite of the prediction received by the corresponding CF example $x'_i$ (i.e., $1 - \hat{y}_{x'_i}$). Intuitively, minimizing $\mathcal{L}_2$ maximizes the validity of the generated CF example $x'_i$ by ensuring that the predictions on $x'_i$ and $x_i$ are different. Finally, the proximity loss $\mathcal{L}_3$ represents the MSE distance between input instance $x_i$ and the CF example $x'_i$, which aims to minimize proximity. This choice of loss functions is crucial to CounterNet's superior performance, as replacing $\mathcal{L}_1$, $\mathcal{L}_2$ and $\mathcal{L}_3$ with alternative functions leads to degraded performance (as we show in Section 4).

### 3.3 TRAINING PROCEDURE

The conventional way of solving the optimization problem in Eq. 1 is to use gradient descent with backpropagation (BP). However, directly optimizing the objective function (Eq. 1) as-is results in two fundamental issues: (1) *poor convergence in training* (shown in Lemma 3.1), and (2) *proneness to adversarial examples* (shown in Lemma 3.2).

**Issue I: Poor Convergence.** Optimizing Eq. 1 as-is via BP leads to poor convergence. This occurs because Eq. 1 contains two different loss objectives with divergent gradients, as Lemma 3.1 shows the gradients of $\mathcal{L}_1$ and $\mathcal{L}_2$ move in opposite directions. Consequently, the accumulated gradient across all three loss objectives fluctuates drastically, which leads to difficulty in training.

**Lemma 3.1** (Divergent Gradient Problem). *Let $\mathcal{L}_1 = \|y - \hat{y}_x\|_2$, and $\mathcal{L}_2 = \|\hat{y}_x - (1 - \hat{y}_{x'})\|_2$, if $x' \to x$, $0 < \hat{y}_x < 1$, $y$ is a binary label, and $|\hat{y}_x - y| < 0.5$, then $\nabla \mathcal{L}_1 \cdot \nabla \mathcal{L}_2 < 0$.*

**Issue II: Adversarial Examples.** Our training procedure should generate high-quality CF examples $x'$ for input instances $x$ without sacrificing the adversarial robustness of the predictor network. Unfortunately, optimizing Eq. 1 as-is is at odds with the goal of achieving adversarial robustness. Lemma 3.2 shows that optimizing $\mathcal{L}_2$ with respect to the predictive weights $\theta_f$ decreases the robustness of the predictor $f(\cdot)$ (by increasing the Lipschitz constant of $f(\cdot)$ (Hein & Andriushchenko, 2017; Wu et al., 2021)), leading to its increased vulnerability to adversarial examples. Proof of Lemma 3.1 and 3.2 can be found in Appendix A.

**Lemma 3.2** (Lipschitz Continuity). *Suppose $f$ is a locally Lipschitz continuous function parameterized by $\theta$, then it satisfies $|f_\theta(x) - f_\theta(x')| \leq K \|x - x'\|_2$, where its Lipschitz constant $K = \sup_{x' \in \mathbb{B}(x, \epsilon)} \{\|\nabla f_\theta(x')\|_2\}$. Let $\mathcal{L}_2 = \|f_\theta(x) - (1 - f_\theta(x'))\|_2$, if $x' \to x$, $0 < f_\theta(\cdot) < 1$, $f(x) \to y$, and $y$ is a binary label, then minimizing $\mathcal{L}_2$ w.r.t. $\theta$ increases the Lipschitz constant $K$.*

We remedy these issues as follows: (1) *to handle poor convergence in training*, we adopt a block-wise coordinate descent procedure, which divides the problem of optimizing Eq. 1 into two parts: (i) optimizing predictive accuracy (primarily influenced by $\mathcal{L}_1$); and (ii) optimizing the validity and proximity of CF generation (primarily influenced by $\mathcal{L}_2$ and $\mathcal{L}_3$). Specifically, for each mini-batch of $m$ data points $\{x^{(i)}, y^{(i)}\}^m$, we apply two gradient updates to the network through backpropagation. For the first update, we compute $\theta^{(1)} = \theta^{(0)} - \nabla_{\theta^{(0)}}(\lambda_1 \cdot \mathcal{L}_1)$, and for the second update, we compute $\theta^{(2)} = \theta^{(1)} - \nabla_{\theta^{(1)}}(\lambda_2 \cdot \mathcal{L}_2 + \lambda_3 \cdot \mathcal{L}_3)$. This procedure ensures that gradients for $\mathcal{L}_1$ and $\mathcal{L}_2$ are calculated separately, which lessens the divergent gradient problem (Lemma 3.1), and leads to significantly better convergence. (2) Moreover, *to improve adversarial robustness* of our predictor network, during the second stage of our coordinate descent procedure (when we optimize for $\lambda_2 \cdot \mathcal{L}_2 + \lambda_3 \cdot \mathcal{L}_3$), we only update the weights in the CF generator $\theta_g$ and freeze gradient updates in both the encoder $\theta_h$ and predictor $\theta_f$ networks. More formally, instead of updating the weights $\theta$ of the entire network during the second update, we only update the CF generator weights $\theta_g$ as follows: $\theta_g^{(2)} = \theta_g^{(1)} - \nabla_{\theta_g^{(1)}}(\lambda_2 \cdot \mathcal{L}_2 + \lambda_3 \cdot \mathcal{L}_3)$. This ensures that the Lipschitz constant of the predictor network does not increase (Lemma 3.2).

## 4 EXPERIMENTAL EVALUATION

We primarily focus our evaluation on heterogeneous tabular datasets for binary classification problems (which is the most common and reasonable setting for CF explanations (Verma et al., 2020; Stepin et al., 2021)). However, CounterNet can be applied to multi-class classification settings, and it can also be adapted to work with other modalities of data, e.g., images, etc. (as shown in Appendix G, I).

**Baselines.** We compare CounterNet against seven state-of-the-art CF explanation methods: (i) *VanillaCF* (Wachter et al., 2017) – which generates CF examples by optimizing CF validity and proximity; (ii) *DiverseCF* (Mothilal et al., 2020), *ProtoCF* (Van Looveren & Klaise, 2019), and *UncertainCF* (Schut et al., 2021) – which optimizes for diversity, consistency with prototypes, and uncertainty, respectively; (iii) *VAE-CF* (Mahajan et al., 2019), *CounteRGAN* (Nemirovsky et al., 2022), *C-CHVAE* (Pawelczyk et al., 2020), and *VCNet* (Guyomard et al., 2022) – which rely on generative models (i.e., VAE or GAN) to generate CF examples [3].

Unlike CounterNet, all of the post-hoc methods require a trained predictive model as input. Thus, for each dataset, we train a neural network model and use it as the target predictive model for all baselines. For a fair comparison, we only keep the encoder and predictor network inside CounterNet's architecture (Figure 2), and optimize them for predictive accuracy alone (i.e., $\mathcal{L}_1$). This combination of encoder and predictor networks is then used as the black-box predictive model for our baselines.

**Datasets.** To remain consistent with prior work on CF explanations (Verma et al., 2020), we evaluate CounterNet on four benchmarked real-world binary classification datasets: (i) *Adult* (Kohavi & Becker, 1996) which predicts whether an individual's income reaches \$50K ($Y=1$) or not ($Y=0$); (ii) *Credit* (Yeh & Lien, 2009) which uses historical payments to predict default of payment ($Y=1$) or not ($Y=0$); (iii) *HELOC* (FICO, 2018) which predicts if a homeowner qualifies for credit ($Y=1$) or not ($Y=0$); (iv) *OULAD* (Kuzilek et al., 2017) which predicts whether MOOC students drop out ($Y=1$) or not ($Y=0$). *We also provide experiments on four additional datasets in Appendix E.*

**Evaluation Metrics.** For each input $x$, CF explanation methods generate two outputs: (i) a prediction $\hat{y}_x$; and (ii) a CF example $x'$. We evaluate the quality of both these outputs using separate metrics. For evaluating predictions, we use *predictive accuracy* (as all four datasets are fairly class-balanced).

For evaluating CF examples, we use five widely used metrics from prior literature (see Appendix B.3 for formal definitions): (i) *Validity* is the fraction of input instances on which CF explanation methods output valid CF examples, i.e., the fraction of input data points for which $\hat{y}_x + \hat{y}_{x'} = 1$. High *validity* is desirable, as it implies the method's effectiveness at creating valid CF examples (Mothilal et al., 2020; Upadhyay et al., 2021). (ii) *Proximity* is the $L_1$ norm distance between $x$ and $x'$ divided by the number of features (Wachter et al., 2017; Mothilal et al., 2020). (iii) *Sparsity* is the number of feature changes between $x$ and $x'$ (normalized by the total number of features) (Wachter et al., 2017; Poursabzi-Sangdeh et al., 2021). Proximity and sparsity serve as proxies for measuring the cost of change of our CF explanation approach, as it is desirable to have fewer modifications in the input space to convert it into a valid CF example. (iv) *Manifold distance* is the $L_1$ distance to the $k$-nearest neighbor of $x'$ (we use $k = 1$ based on (Verma et al., 2022)). Low *manifold distance* is desirable as closeness to the training data manifold indicates realistic CF explanations (Van Looveren & Klaise, 2019; Verma et al., 2022). (v) Finally, we also report the *runtime* for generating CF examples.

### 4.1 EVALUATION OF COUNTERNET PERFORMANCE

**Predictive Accuracy.** Table 1 compares CounterNet's predictive accuracy against the base prediction model used by baselines. This table shows that CounterNet exhibits highly competitive predictive performance - it achieves marginally better accuracy on the Credit dataset (row 2), and achieves marginally lower accuracy on the remaining datasets. Across all four datasets, the difference between the predictive accuracy of Counter-Net and the base model is $\sim 0.1\%$. Thus, the potential

Table 1: Predictive Accuracy of CounterNet

| Dataset | Base Model | CounterNet |
|---------|-----------|-----------|
| Adult   | 0.831     | 0.828     |
| Credit  | 0.813     | 0.819     |
| HELOC   | 0.717     | 0.716     |
| OULAD   | 0.934     | 0.929     |

---

[3]Note that Yang et al. (2021) propose another parametric post-hoc method, but we exclude it in our baseline comparison because it achieves comparable performance to C-CHVAE (as reported in (Yang et al., 2021)).

Table 2: Evaluation of CF explanations: CounterNet achieves perfect validity (i.e., Val.), and it incurs comparable (or lesser) cost of changes (i.e., Prox, Spar.) than baseline methods, with comparable manifold distance (i.e., Man.). **Bold** and *italicized* cells highlight the best and second-best performing methods, respectively.

| Method | Adult | | | | Credit | | | | HELOC | | | | OULAD | | | |
|---|---|---|---|---|---|---|---|---|---|---|---|---|---|---|---|---|
| | Val. | Prox. | Spar. | Man. | Val. | Prox. | Spar. | Man. | Val. | Prox. | Spar. | Man. | Val. | Prox. | Spar. | Man. |
| VanillaCF | 0.76 | .202 | **.556** | *0.57* | 0.92 | **.123** | .841 | 0.59 | **1.00** | .154 | .883 | 0.71 | **1.00** | .101 | .762 | 1.30 |
| DiverseCF | 0.54 | .276 | .662 | 1.16 | **1.00** | .264 | .918 | 1.68 | 0.90 | .149 | *.434* | 1.34 | 0.68 | .117 | **.565** | 2.51 |
| ProtoCF | 0.59 | .250 | .648 | 0.62 | 0.92 | .197 | .855 | 0.82 | **1.00** | .168 | .805 | *0.56* | **1.00** | .107 | .754 | 1.46 |
| UncertainCF | 0.36 | .307 | .713 | 1.23 | 0.62 | .155 | **.217** | 0.80 | 0.55 | .130 | **.161** | 0.94 | 0.59 | .098 | .734 | 2.23 |
| C-CHVAE | **1.00** | .281 | .721 | 0.94 | **1.00** | .357 | .853 | 1.85 | **1.00** | .155 | .790 | 0.81 | **1.00** | .110 | .797 | 2.11 |
| VAE-CF | 0.66 | .287 | .734 | 1.03 | 0.13 | .201 | .756 | 0.62 | **1.00** | .221 | .893 | 1.04 | **1.00** | .115 | .586 | 2.19 |
| CounteRGAN | 0.78 | .327 | .698 | 2.21 | 0.39 | .260 | .687 | 2.03 | **1.00** | .271 | .509 | 2.23 | 0.43 | .087 | .587 | 2.15 |
| VCNet | **1.00** | .291 | .755 | **0.19** | **1.00** | .162 | .939 | **0.16** | **1.00** | .154 | .786 | **0.39** | **1.00** | .095 | .903 | 1.33 |
| CounterNet | **1.00** | **.196** | *.644* | 0.64 | **1.00** | *.132* | .912 | *0.56* | **1.00** | **.125** | .740 | *0.56* | **1.00** | **.075** | .725 | **0.87** |

benefits achieved by CounterNet's joint training of predictor and CF generator networks do not come at a cost of reduced predictive accuracy.

**Counterfactual Validity.** Table 2 compares the validity achieved by CounterNet and baselines on all four datasets. We observe that CounterNet, C-CHVAE, and VCNet are the only three methods with 100% validity on all datasets. With respect to the other baselines, CounterNet achieves 8% and 12.3% higher average validity (across all datasets) than VanillaCF and ProtoCF (our next best baselines).

**Proximity & Sparsity.** Table 2 compares the proximity/sparsity achieved by all methods. CounterNet achieves at least 3% better proximity than all other baselines on three out of four datasets (Adult, HELOC, and OULAD), and it is the second best performing model on the Credit dataset (where it achieves 7.3% poorer proximity than VanillaCF). In terms of sparsity, CounterNet performs reasonably well, it is the second best performing model on the Adult and HELOC datasets even though CounterNet does not explicitly optimize for sparsity. *This shows that CounterNet outperforms all baselines by generating CF examples with the highest validity and best proximity scores.*

**Cost-Invalidity Trade-off.** We illustrate the cost-invalidity trade-off (Rawal et al., 2020) for all methods. Figure 3 shows that CounterNet lies on the bottom left of this figure — it consistently achieves the lowest invalidity and cost on all four datasets. In comparison, VCNet achieves the same perfect validity, but at the expense of ∼34% higher cost than CounterNet. Similarly, C-CHVAE demands ∼71% higher cost than CounterNet to achieve perfect validity. On the other hand, VanillaCF achieves comparable cost to CounterNet (10% higher cost), but it achieves lower validity by 8%. This shows that CounterNet's joint training enables it to properly balance the cost-invalidity trade-off.

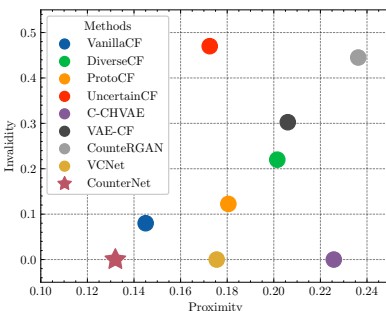

Figure 3: Illustration of the cost-invalidity trade-off across all four datasets. Methods at the bottom left are preferable.

Table 3: Runtime comparison (in milliseconds). CounterNet outperforms all of baselines in runtime.

| Method | Adult | Credit | HELOC | OULAD |
|---|---|---|---|---|
| VanillaCF | 1432.09 | 1358.26 | 1340.42 | 1705.93 |
| DiverseCF | 4685.39 | 3898.43 | 3921.72 | 5478.17 |
| ProtoCF | 2348.21 | 2056.01 | 1956.71 | 2823.29 |
| UncertainCF | 379.95 | 60.80 | 7.91 | 6.81 |
| C-CHVAE | 3.28 | 568.28 | 2.68 | 4.79 |
| VAE-CF | 1.72 | 1.28 | 1.48 | 1.84 |
| CounteRGAN | 1.96 | 1.77 | 1.59 | 2.40 |
| VCNet | 1.39 | 1.23 | 1.13 | 1.81 |
| CounterNet | **0.64** | **0.39** | **0.44** | **0.79** |

Table 4: Ablation analysis of CounterNet. Each ablation leads to degraded performance, which in turn, demonstrates the importance of different design choices inside CounterNet.

| Ablation | Adult | | Credit | | HELOC | | OULAD | |
|---|---|---|---|---|---|---|---|---|
| | Val. | Prox. | Val. | Prox. | Val. | Prox. | Val. | Prox. |
| CounterNet-BCE | 0.86 | .238 | 0.96 | .210 | 0.86 | .238 | 0.95 | .101 |
| CounterNet-SingleBP | 0.64 | .248 | 0.92 | .251 | 0.93 | .206 | 0.94 | .110 |
| CounterNet-Separate | 0.96 | .257 | 0.99 | .265 | 0.91 | .161 | 0.94 | .097 |
| CounterNet-NoPass-$p_x$ | 0.97 | .256 | 0.99 | .339 | 0.98 | .147 | 0.98 | .101 |
| CounterNet-Posthoc | 1.00 | .276 | 1.00 | .247 | 1.00 | .153 | 0.99 | .099 |
| CounterNet | **1.00** | **.196** | **1.00** | **.132** | **1.00** | **.125** | **1.00** | **.075** |

**Manifold Distance.** Table 2 shows that CounterNet achieves the second-lowest manifold distance in average (right below VCNet, which explicitly optimizes for data manifold). In particular, CounterNet achieves the lowest manifold distance in *OULAD*, and is ranked second in *Credit* and *HELOC*. This result shows that CounterNet generates highly realistic CF examples that adhere to the data manifold.

**Running Time.** Table 3 shows the average runtime (in milliseconds) of different methods to generate a CF example for a single data point. CounterNet outperforms all seven baselines in every dataset. In particular, CounterNet generates CF examples ~3X faster than VAE-CF, CouneRGAN, and VCNet, ~5X faster than C-CHVAE, and three orders of magnitude (>1000X) faster than other baselines. This result shows that CounterNet is more usable for adoption in time-constrained environments.

## 4.2 FURTHER ANALYSIS

**Ablation Analysis.** We analyze five ablations of CounterNet to underscore the design choices inside CounterNet. First, we accentuate the importance of the MSE loss functions used to optimize CounterNet (Eq. 1) by replacing the MSE based $\mathcal{L}_1$ and $\mathcal{L}_2$ loss in Eq. 1 with binary cross entropy loss (*CounterNet-BCE*). Second, we underscore the importance of CounterNet's two-stage coordinate descent procedure by using conventional one-step BP optimization to train CounterNet instead (*CounterNet-SingleBP*). In addition, we validate CounterNet's architecture design by experimenting two alternative designs: (i) we use a *separate* predictor $f : \mathcal{X} \to \mathcal{Y}$ and CF generator $g : \mathcal{X} \to \mathcal{X}'$, such that $f$ and $g$ share no identical components (unlike in CounterNet, where $z_x$ are shared with both $f$ and $g$; *CounterNet-Separate*); and (ii) we highlight the design choice of passing $p_x$ to the CF generator by excluding passing $p_x$ (*CounterNet-NoPass-$p_x$*). Finally, we highlight the importance of the joint-training of predictor and CF generator in CounterNet by training the CounterNet in a post-hoc fashion (*CounterNet-Posthoc*), i.e., we first train the predictor on the *entire* training dataset, and optimize CF generator while the trained predictor is frozen.

Table 4 compares the validity and proximity achieved by CounterNet and five ablations. Importantly, each ablation leads to degraded performance as compared to CounterNet, which demonstrates CounterNet's different design choices. *CounterNet-BCE* and *CounterNet-SingleBP* perform poorly in comparison, which illustrates the importance of the MSE-based loss function and block-wise coordinate descent procedure. Similarly, *CounterNet-Separate* and *CounterNet-NoPass-$p_x$* achieve degraded validity and proximity scores, which highlight the importance of CounterNet's architecture design. Finally, *CounterNet-Posthoc* achieves comparable validity as CounterNet, but fails to match the performance of proximity. This result demonstrates the importance of the joint-training procedure of CounterNet in optimally balancing the cost-invalidity trade-off.

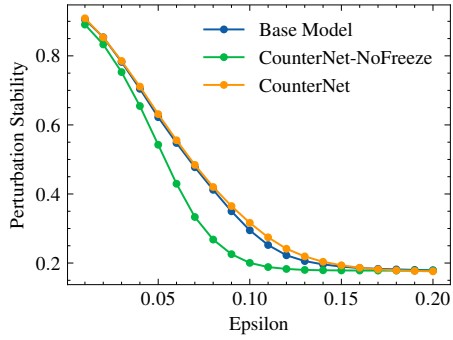

Figure 4: Robustness of the predictor $f(\cdot)$. CounterNet reaches the upper bound of robustness (i.e., comparable to the base model).

**Adversarial Robustness.** We illustrate that CounterNet does not suffer from decreased robustness of the predictor network resulting from optimizing for the validity loss $\mathcal{L}_2$ (as shown in Lemma 3.2). We compare the robustness of CounterNet's predictor network $f(\cdot)$ against two baselines: (i) the base predictive model described in Table 1; and (ii) CounterNet without freezing the predictor at the second stage of our coordinate descent optimization (*CounterNet-NoFreeze*). Figure 4 illustrates the perturbation stability (Wu et al., 2021) of all three CounterNet variants against adversarial examples (generated via projected gradient descent (Madry et al., 2018)). CounterNet achieves comparable perturbation stability as the base model, which indicates that CounterNet reaches its robustness upper bound (i.e., the robustness of the base model). Moreover, the empirical results in Figure 4 confirm Lemma 3.2 as *CounterNet-NoFreeze* achieves significantly poorer stability. We observe similar patterns with different attack methods on other datasets (see Appendix C.1). These results show that by freezing the predictor and encoder networks at the second stage of our coordinate descent procedure, CounterNet suffers less from the vulnerability issue created by the adversarial examples.

**Feasibility of CF Explanations.** Finally, we show how CounterNet's training procedure can be adapted to ensure that the generated CF examples are feasible. In particular, we attempt to use projected gradient descent during the training of CounterNet and enforce hard constraints during the inference stage in order to ensure that the generated CF examples satisfy immutable feature constraints (e.g., `gender` should remain unchanged). At the training stage, a CF example $x'$ is first generated from $g(\cdot)$, and is projected into its feasible space (i.e., $x'' = \mathbb{P}(x')$). Next, we optimize CounterNet over the prediction $\hat{y}_x$ and its projected CF example $x''$ (via our block-wise coordinate descent procedure). During inference, we enforce that the set of immutable features remains unchanged. Table 5 shows that enforcing immutable features (via projected gradient descent) does not negatively impact the validity and proximity of the CF examples. This result shows that CounterNet can produce CF examples that respect feasibility constraints.

Table 5: Impact of the immutable feature constraints in CounterNet. CounterNet generates feasible CF explanations without sacrificing validity and proximity.

| Dataset | Val. Diff. | Prox. Diff. |
|---------|:----------:|:-----------:|
| Adult   | 0.0        | .009        |
| Credit  | 0.0        | .005        |
| OULAD   | 0.0        | .004        |

## 5 DISCUSSION & CONCLUSION

Although our experiments exhibit CounterNet's superior performance than post-hoc baselines, these two methods have somewhat different motivations. While post-hoc methods are designed for generating CF explanations for trained black-box ML models (whose training data and model weights might not be available), CounterNet is most suitable when the ML model developers aspire to build prediction and explanation module from scratch, where the training data can be exploited to optimize the generation of CF examples. We anticipate CounterNet to be valuable for service providers who wish to comply with GDPR-style regulations without sacrificing their operational effectiveness (e.g., reduced predictive power). Importantly, CounterNet can still be used to interpret proprietary ML models by forcing its predictor network to mimic that proprietary model (see Appendix D.1).

CounterNet has two limitations: (i) We do not consider other desirable aspects in CF explanations, such as diversity (Mothilal et al., 2020), recourse cost (Ustun et al., 2019), fairness (Von Kügelgen et al., 2022), and causality (Karimi et al., 2021). Further research is needed to address these issues. (ii) It is also important to ensure that generated CF examples do not amplify or provide support to the narratives resulting from pre-existing race-based and gender-based societal inequities (among others). One short-term workaround is to have humans in the loop. We can provide CounterNet's explanations as a decision-aid to a well-trained human official, who is in charge of communicating the decisions of ML models to human end-users in a respectful and humane manner. In the long-run, further qualitative studies are needed to understand the social impacts of CounterNet.

This paper proposes *CounterNet*, which integrates predictive model training and CF example generation into a single *end-to-end* pipeline. Unlike prior work, CounterNet ensures that the objectives of predictive model training and CF example generation are closely aligned. We adopt a block-wise coordinate descent procedure to effectively train CounterNet. Experimental results show that CounterNet outperforms state-of-the-art baselines in validity, proximity, and runtime, and is highly competitive in predictive accuracy, sparsity, and closeness to data manifold.

## 6 ETHICS & REPRODUCIBILITY STATEMENT

**Ethics Statement.** Although CounterNet is suitable for real-time deployment given its superior performance in its highly aligned CF explanations and speed, one must be aware of the possible negative impacts of its CF explanations to human end-users. It is important to ensure that generated CF examples do not amplify or provide support to the narratives resulting from pre-existing race-based and gender-based societal inequities (among others). As we stated in Section 5, one short-term workaround is to have humans in the loop. We can provide CounterNet's explanations as a decision-aid to a well-trained human official, who is in charge of communicating the decisions of ML models to human end-users in a respectful and humane manner. In the long-run, further qualitative and quantitative studies are needed to understand the social impacts of CounterNet.

**Reproducibility Statement.** To aid the reproducibility of this work, we provide the code in the supplement material, and will make the code public once it is accepted. We also provide the dataset used for evaluating this paper in this anonymous repository. In addition, we outline the choices of hyperparameters in Appendix B.4. A detailed description of our experimental implementation can be found in Appendix B. For theoretical analysis of our novel block-wise coordinate descent procedure, we provide complete proof in Appendix A.

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

# Appendix

# A SUPPLEMENTAL PROOF

## A.1 PROOF OF LEMMA 3.1

*Proof.* $\nabla_\theta \mathcal{L}_1 = \nabla_\theta \|y - \hat{y}_x\|_2$, and $\nabla_\theta \mathcal{L}_2 = \nabla_\theta \|(1 - \hat{y}_x) - \hat{y}_{x'}\|_2$. Then, we have

$$
\begin{aligned}
\nabla_\theta \mathcal{L}_1 &= \nabla_\theta y^2 - 2y \cdot \nabla_\theta \cdot \hat{y}_x + \nabla_\theta \hat{y}_x^2 \\
&= -2y \cdot \nabla_\theta \hat{y}_x + \nabla_\theta \hat{y}_x^2 \\
&= -2y \cdot \nabla_\theta \hat{y}_x + 2\hat{y}_x \nabla_\theta \hat{y}_x \\
&= 2(\hat{y}_x - y) \cdot \nabla_\theta \hat{y}_x
\end{aligned}
$$

Since $x' \to x$, as we expect CF example $x'$ is closed to the original instance x, we can replace $x'$ to $x$ in $\mathcal{L}_2$. Then, we have

$$
\begin{aligned}
\nabla_\theta \mathcal{L}_2 &= \nabla_\theta (1 - 2\hat{y}_x)^2 \\
&= -4 \cdot \nabla_\theta \hat{y}_x + 4 \cdot \nabla_\theta \hat{y}_x^2 \\
&= -4 \cdot \nabla_\theta \hat{y}_x + 4 \cdot 2 \cdot \hat{y}_x \cdot \nabla_\theta \hat{y}_x \\
&= 4 \cdot (2\hat{y}_x - 1) \nabla_\theta \hat{y}_x
\end{aligned}
$$

Hence,

$$
\begin{aligned}
\nabla_\theta \mathcal{L}_1 \cdot \nabla_\theta \mathcal{L}_2 &= 2 \cdot (\hat{y}_x - y) \cdot \nabla_\theta \hat{y}_x \cdot 4 \cdot (2\hat{y}_x - 1) \nabla_\theta \hat{y}_x \\
&= 8 \cdot (\hat{y}_x - y) \cdot (2\hat{y}_x - 1)(\nabla_\theta \hat{y}_x)^2
\end{aligned}
$$

Since $(\nabla_\theta \hat{y}_x)^2 > 0$, we only need to prove whether $(\hat{y}_x - y) \cdot (2\hat{y}_x - 1)$ is positive or negative. Given that $|\hat{y}_x - y| < 0.5$,

- if $y = 1$, we have $0.5 < \hat{y}_x < 1$. Then, $(\hat{y}_x - y) < 0$, $(2\hat{y}_x - 1) > 0$.

- if $y = 0$, we have $0 < \hat{y}_x < 0.5$. Then, $(\hat{y}_x - y) > 0$, $(2\hat{y}_x - 1) < 0$.

Therefore, $(\hat{y}_x - y) \cdot (2\hat{y}_x - 1) < 0$. Hence, $\nabla_\theta \mathcal{L}_1 \cdot \nabla_\theta \mathcal{L}_2 < 0$.

$\square$

## A.2 PROOF OF LEMMA 3.2

*Proof.* Assuming $f_\theta(x) \to y$ as we expect the predictor network produces accurate predictions, and $y = \{0, 1\}$, we can replace $f_\theta(x)$ to $y$. Then, minimizing $\mathcal{L}_2$ (in Lemma 3.2) indicates minimizing $\|y - (1 - f_\theta(x'))\|_2$. Since $0 < f_\theta(\cdot) < 1$, we have

$$
\min \|y - (1 - f_\theta(x'))\|_2 = \max \|y - f_\theta(x')\|_2
$$

By replacing $y$ to $f_\theta(x)$, then minimizing $\mathcal{L}_2$ indicates maximizing $\|f_\theta(x) - f_\theta(x')\|_2$. By definition, the lipschitz constant $K$ is

$$
K = \sup_{x' \in \mathbb{B}(x,\epsilon)} \{\|\nabla f_\theta(x')\|_2 = \sup_{x' \in \mathbb{B}(x,\epsilon)} \left\{ \frac{\|f_\theta(x) - f_\theta(x')\|}{\|x - x'\|} \right\}
$$

where minimizing $\mathcal{L}_2$ increases $\|f_\theta(x) - f_\theta(x')\|_2$. Therefore, the lipschitz constant $K$ increases.

$\square$

# B  IMPLEMENTATION DETAILS

Here we provide implementation details of CounterNet and five baselines on four datasets listed in Section 4. The code can be found in the supplemental material.

## B.1  SOFTWARE AND HARDWARE SPECIFICATION

We use Python (v3.7) with Pytorch (v1.82), Pytorch Lightning (v1.10), numpy (v1.19.3), pandas (1.1.1) and scikit-learn (0.23.2) for the implementations. All our experiments were run on a Debian-10 Linux-based Deep Learning Image with CUDA 11.0 on the Google Cloud Platform.

The CounterNet' network is trained on NVIDIA Tesla V100 with an 8-core Intel machine. CF generation of four baselines are run on a 16-core Intel machine with 64 GB of RAM. The evaluation are generated from the same 16-core machine.

## B.2  DATASETS FOR EVALUATION

Here, we reiterate our used datasets for evaluations. Our evaluation is conducted on eight widely-used tabular datasets. Our primary evaluation uses four large-sized datasets (shown in Section 4), including *Adult*, *Credit*, *HELOC*, and *OULAD*, which contain at least 10k data instances. In addition, we experiment with four small-sized datasets, including *Student*, *Titanic*, *Cancer*, and *German*. Table 6 summarizes datasets used for evaluations.

Table 6: Summary of Datasets used for Evaluation

| Dataset | Size | #Continuous | #Categorical |
|---------|------|-------------|--------------|
| Adult   | 32,561 | 2  | 6  |
| Credit  | 30,000 | 20 | 3  |
| HELOC   | 10,459 | 21 | 2  |
| OULAD   | 32,593 | 23 | 8  |
| Student | 649    | 2  | 14 |
| Titanic | 891    | 2  | 24 |
| Cancer  | 569    | 30 | 0  |
| German  | 1,000  | 7  | 13 |

## B.3  EVALUATION METRICS

Here, we provide formal definitions of the evaluation metrics.

**Predictive Accuracy** is defined as the fraction of the correct predictions.

$$\texttt{Predictive-Accuracy} = \frac{\#|f(x) = y|}{n} \qquad (2)$$

**Validity** is defined as the fraction of input instances on which CF explanation methods output valid CF examples.

$$\texttt{Validity} = \frac{\#|f(x') = 1 - y|}{n} \qquad (3)$$

**Proximity** is defined as the $L_1$ norm distance between $x$ and $x'$ divided by the number of features.

$$\texttt{Proximity} = \frac{1}{nd} \sum_{i=1}^{n} \sum_{j=1}^{d} \|x_i^{(j)} - x_i'^{(j)}\|_1 \qquad (4)$$

**Sparsity** is defined as the fraction of the number of feature changes between $x$ and $x'$.

$$\texttt{Sparsity} = \frac{1}{nd} \sum_{i=1}^{n} \sum_{j=1}^{d} \|x_i^{(j)} - x_i'^{(j)}\|_0 \qquad (5)$$

## B.4  COUNTERNET IMPLEMENTATION DETAILS

Across all six datasets, we apply the following same settings in training CounterNet: We initialize the weights as in He et al. (2016). We adopt the Adam with mini-batch size of 128. For each datasets,

we trained the models for up to $1 \times 10^3$ iterations. To avoid gradient explosion, we apply gradient clipping by setting the threshold to 0.5 to clip gradients with norm above 0.5. We set dropout rate to 0.3 to prevent overfitting. For all six datasets, we set $\lambda_1 = 1.0$, $\lambda_2 = 0.2$, $\lambda_3 = 0.1$ in Equation 1.

The learning rate is the only hyper-parameter that varies across six datasets. From our empirical study, we find the training to CounterNet is sensitive to the learning rate, although a good choice of loss function (e.g. choosing MSE over cross-entropy) can widen the range of an "optimal" learning rate. We apply grid search to tune the learning rate, and our choice is specified in Table 7.

Additionally, we specify the architecture's details (e.g. dimensions of each layer in encoder, predictor and CF generator) in Table 7. The numbers in each bracket represent the dimension of the transformed matrix. For example, the encoder dimensions for adult dataset is [29, 50, 10], which means that the dimension of input $x \in \mathbb{R}^d$ is 29 (e.g. $d = 29$); the encoder first transforms the input into a 50 dimension matrix, and then downsamples it to generate the latent representation $z \in \mathbb{R}^k$ where $k = 10$.

Table 7: Hyperparameters and architectures for each dataset.

| Dataset | Learning Rate | Encoder Dims | Predictor Dims | CF Generator Dims |
|---------|---------------|--------------|----------------|-------------------|
| Adult   | 0.003 | [29, 50, 10]   | [10, 10, 2] | [20, 50, 29]   |
| Credit  | 0.003 | [33, 50, 10]   | [10, 10, 2] | [20, 50, 33]   |
| HELOC   | 0.005 | [35, 100, 10]  | [10, 10, 2] | [20, 100, 35]  |
| OULAD   | 0.001 | [127, 200, 10] | [10, 10, 2] | [20, 200, 127] |
| Student | 0.01  | [85, 100, 10]  | [10, 10, 2] | [20, 100, 85]  |
| Titanic | 0.01  | [57, 100, 10]  | [10, 10, 2] | [20, 100, 57]  |
| Cancer  | 0.001 | [30, 50, 10]   | [10, 10, 2] | [20, 50, 30]   |
| German  | 0.003 | [61, 50, 10]   | [10, 10, 2] | [20, 50, 61]   |

### B.5 HYPER-PARAMETERS FOR BASELINES

Next, we describe the implementation of baseline methods. For VanillaCF and ProtoCF, we follow author's instruction as much as we can, and implement them in Pytorch. For VanillaCF, DiverseCF and ProtoCF, we run maximum $1 \times 10^3$ steps. After CF generation, we convert the results to one-hot-encoding format for each categorical feature. For training the VAE-CF, we follow Mahajan et al. (2019)'s settings on running maximum 50 epoches and setting the batch size to 1024. We use the same learning rate as in Table 7 for VAE training.

Table 8: Learning rate of the base predictive models on each dataset.

| Dataset | Learning Rate |
|---------|---------------|
| Adult   | 0.01  |
| HELOC   | 0.005 |
| OULAD   | 0.001 |
| Student | 0.01  |
| Titanic | 0.01  |
| Cancer  | 0.001 |

For training predictive models for baseline algorithms, we apply grid search for tuning the learning rate, which is specified in Table 8. Similar to training the CounterNet, we adopt the Adam with mini-batch size of 128, and set the dropout rate to 0.3. We train the model for up to 100 iterations with early stopping to avoid overfittings.

## C  ADDITIONAL EXPERIMENTAL RESULTS

Here, we provide additional results of experiments in Section 4. These results further demonstrate the effectiveness of CounteNet.

### C.1  ADDITIONAL ROBUSTNESS RESULTS

We provide supplementary results on evaluating the robustness of the predictor network on three large datasets (i.e., Adult, HELOC and OULAD). In particular, we implement FSGM (Goodfellow et al., 2015) and PGD (Madry et al., 2018) attack for testing the robustness of the predictive models. Figure 5 illustrates that CounterNet achieves comparable perturbation stability (i.e., the robustness of

the predictive model) as the base model. In addition, Figure 5 supports the findings in Lemma 3.2 since *CounterNet-NoFreeze* consistently achieves lower stability than base models and CounterNet.

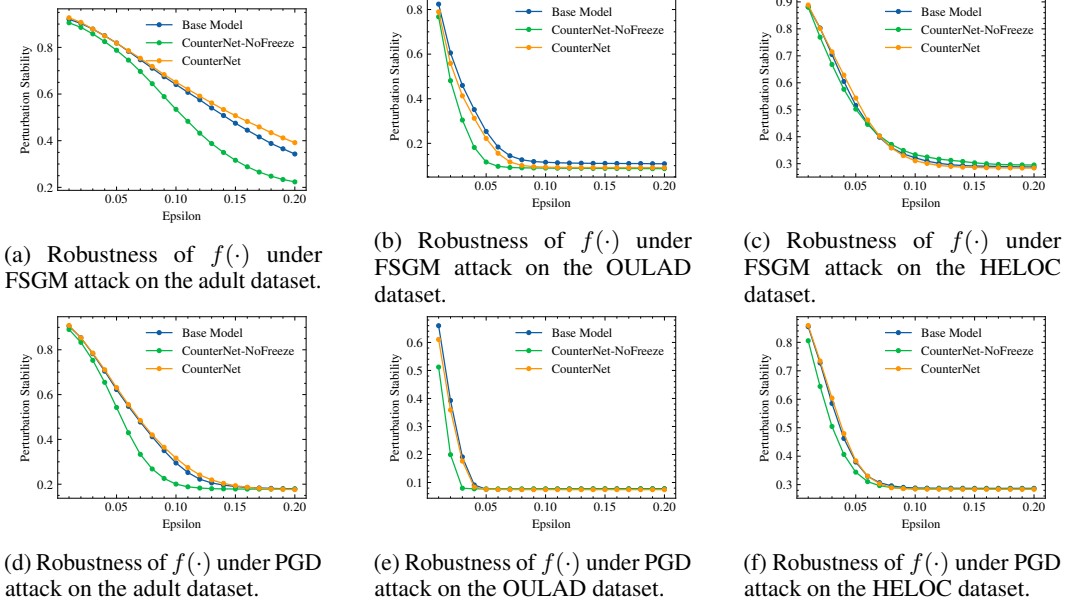

(a) Robustness of $f(\cdot)$ under FSGM attack on the adult dataset.

(b) Robustness of $f(\cdot)$ under FSGM attack on the OULAD dataset.

(c) Robustness of $f(\cdot)$ under FSGM attack on the HELOC dataset.

(d) Robustness of $f(\cdot)$ under PGD attack on the adult dataset.

(e) Robustness of $f(\cdot)$ under PGD attack on the OULAD dataset.

(f) Robustness of $f(\cdot)$ under PGD attack on the HELOC dataset.

Figure 5: Robustness of $f(\cdot)$ under FSGM attack (5a-5c) (Goodfellow et al., 2015) and PGD (5d-5f) (Madry et al., 2018) attack.

## C.2 TRAINING TIME OF COUNTERNET

Table 9 shows the training time of base model and CounterNet for each epoch in seconds. CounterNet takes roughly 3X times training time as compared to the base model (which is only trained for the predictive performance). Note that the training time is a secondary metric for evaluating the speed, as the training of the model only occurs once, whereas the inference time (i.e., runtime in Table 2) is a more important metric, as it keeps increasing during the deployment stage.

Table 9: Training time of base model and CounterNet for each epoch (in second).

| Dataset | Base Model | CounterNet |
|---------|------------|------------|
| Adult   | 0.73       | 2.11       |
| HELOC   | 0.22       | 0.65       |
| OULAD   | 0.72       | 2.11       |

## D ADDITIONAL ABLATION STUDY

### D.1 COUNTERNET UNDER THE BLACK-BOX ASSUMPTIONS

We illustrate how CounterNet can be adapted to the post-hoc black-box setting. In this setting, CF explanation methods generate CF explanations for a *trained* black-box model (with access to the model's output). CounterNet can also be used in this post-hoc setting by forcing the predictor network to surrogate the black-box model. Specifically, let a black-box model $M : \mathcal{X} \rightarrow \mathcal{Y}$ outputs the predictions, our goal of training the predictor is to ensure that the predictor model behaves like the black-box model (i.e., $\mathcal{M}(x) = f(x)$). The training objective of CounterNet is

$$\underset{\theta}{\operatorname{argmin}} \frac{1}{N} \sum_{i=1}^{N} \left[ \lambda_1 \cdot \underbrace{(M(x_i) - \hat{y}_{x_i})^2}_{\text{Prediction Loss } (\mathcal{L}_1)} + \lambda_2 \cdot \underbrace{\left(\hat{y}_{x_i} - \left(1 - \hat{y}_{x'_i}\right)\right)^2}_{\text{Validity Loss } (\mathcal{L}_2)} + \lambda_3 \cdot \underbrace{(x_i - x'_i)^2}_{\text{Proximity Loss } (\mathcal{L}_3)} \right] \quad (6)$$

Table 10: Evaluation of CounterNet under the post-hoc setting. CFNET-BB represents the CounterNet evaluated under the *black-box* setting. CFNET-PH represents the CounterNet trained via a *post-hoc* fashion, which in turn, demonstrates the importance of joint-training procedure in CounterNet.

| Method | Adult | | | | Credit | | | | HELOC | | | | OULAD | | | |
|---|---|---|---|---|---|---|---|---|---|---|---|---|---|---|---|---|
| | Val. | Prox. | Spar. | Man. | Val. | Prox. | Spar. | Man. | Val. | Prox. | Spar. | Man. | Val. | Prox. | Spar. | Man. |
| CFNET-BB | 0.99 | .217 | .716 | 0.73 | .99 | .138 | .861 | 0.64 | 0.98 | .158 | .758 | 0.58 | 0.99 | .073 | .641 | 0.96 |
| CFNET-PH | 1.00 | .276 | .663 | 1.26 | 1.00 | .247 | .804 | 1.36 | 1.00 | .153 | .815 | 0.83 | 0.99 | .099 | .731 | 1.64 |
| CounterNet | 1.00 | .196 | .644 | 0.64 | 1.00 | .132 | .912 | 0.56 | 1.00 | .125 | .740 | 0.56 | 1.00 | .075 | .725 | 0.87 |

Table 11: Ablation analysis of CounterNet. Each ablation leads to degraded performance, which in turn, demonstrates the importance of different design choices inside CounterNet.

| Ablation | Adult | | Credit | | HELOC | | OULAD | |
|---|---|---|---|---|---|---|---|---|
| | Val. | Prox. | Val. | Prox. | Val. | Prox. | Val. | Prox. |
| CounterNet-$l_1$ | .98 | 0.25 | 0.99 | .163 | .99 | .155 | 0.99 | .094 |
| CounterNet | 1.00 | .196 | 1.00 | .132 | 1.00 | .125 | 1.00 | .075 |

Note that Eq. 6 looks identical to Eq. 1. The only difference is that $y_i$ in Eq. 1 is replaced to $M(x_i)$.

Table 10 shows the performance of CounterNet under the black-box setting (CFNET-BB). CFNET-BB degrades slightly in terms of validity, average $L_1$ to *CounterNet*. This is because approximating the black-box model leads to degraded performance in the quality of generating CF explanations.

## D.2    ABLATIONS ON COUNTERNET'S TRAINING

In addition, we provide supplementary results on ablation analysis of three large datasets (Adult, HELOC, and OULAD) to understand the design choices of the CounterNet training, shown in Figure 6). This figure shows that compared to CounterNet's learning curve for $\mathcal{L}_2$, *CounterNet-BCE* and *CounterNet-NoSmooth*'s learning curves show significantly higher instability, illustrating the importance of MSE-based loss functions and label smoothing techniques. Moreover, *CounterNet-SingleBP*'s learning curve for $\mathcal{L}_2$ performs poorly in comparison, which illustrates the difficulty of optimizing three divergent objectives using a single BP procedure. In turn, this also illustrates the effectiveness of our block-wise coordinate descent optimization procedure in CounterNet's training. These results show that all design choices made in Section 3 contribute to training the model effectively.

In addition, we experiment with alternative loss formulations. We replace the MSE based $\mathcal{L}_3$ loss in Eq. 1 with $l_1$ norm (CounterNet-$l_1$). Table 11 shows that replacing $\mathcal{L}_3$ with a $l_1$ formulation leads to a degraded performance.

## E    EXPERIMENTAL EVALUATION ON SMALL-SIZED DATASETS

In addition to four large datasets in Section 4, we experiment with four small-sized datasets: (i) *Breast Cancer Wisconsin* (Blake, 1998) which classifies malignant (Y=1) or benign (Y=0) tumors; (ii) *Student Performance* (Cortez & Silva, 2008) which predicts whether a student will pass (Y=1) or fail (Y=0) the exam; (iii) *Titanic* (Kaggle, 2018) which predicts whether passengers survived (Y=1) the Titanic shipwreck or not (Y=0); and (iv) *German Credit* (Asuncion & Newman, 2007) which predicts whether the credit score of a customer is good (Y=1) or bad (Y=0).

Table 12 compares the validity, average $L_1$ and sparsity achieved by CounterNet and baselines. Similar to results in Table 2, CounterNet achieves a perfect validity. In addition, CounterNet achieves the lowest proximity in three out of four small datasets. This result further shows CounterNet's ability in balancing the cost-invalidity trade-off.

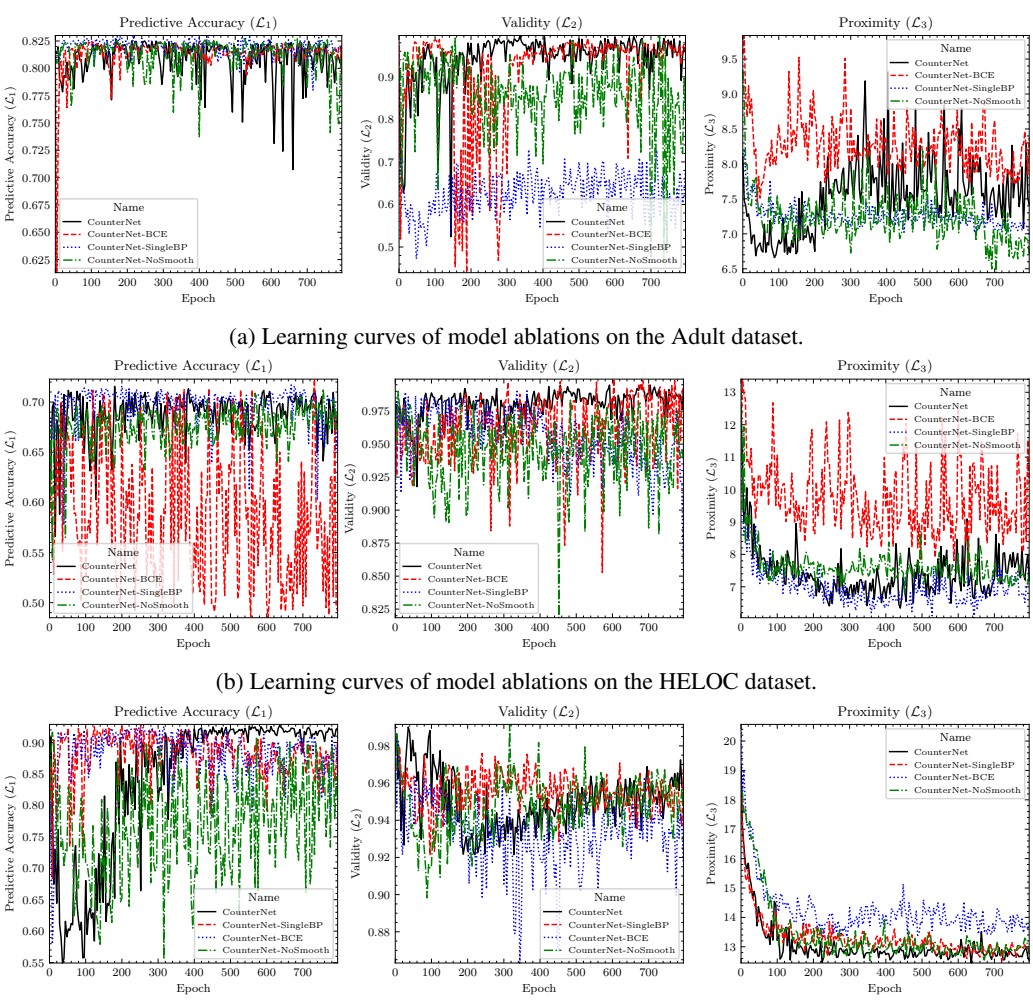

(a) Learning curves of model ablations on the Adult dataset.

(b) Learning curves of model ablations on the HELOC dataset.

(c) Learning curves of model ablations on the OULAD dataset.

Figure 6: Learning curves of $\mathcal{L}_1$ (left), $\mathcal{L}_2$ (mid), and $\mathcal{L}_3$ (right) of model ablations on the Adult (6a), HELOC (6b), and OULAD (6c) dataset.

Table 12: Evaluation of counterfactual explanations on four small-sized datasets.

| Method | Student | | | Titanic | | | Cancer | | | German | | |
|---|---|---|---|---|---|---|---|---|---|---|---|---|
| | Val. | Prox. | Spar. | Val. | Prox. | Spar. | Val. | Prox. | Spar. | Val. | Prox. | Spar. |
| VanillaCF | 0.80 | 0.101 | 0.762 | 0.91 | 0.289 | 0.381 | **1.00** | 0.135 | 0.278 | 0.86 | 0.384 | 0.967 |
| DiverseCF | 0.53 | 0.117 | 0.565 | 0.52 | 0.321 | 0.370 | 0.99 | 0.075 | 0.157 | 0.64 | 0.246 | 1.000 |
| ProtoCF | 0.32 | 0.107 | 0.754 | 0.76 | 0.305 | 0.383 | **1.00** | 0.070 | 0.167 | 0.82 | 0.369 | 0.983 |
| UncertainCF | 0.45 | 0.251 | 0.675 | 0.41 | 0.422 | 0.512 | **1.00** | **0.023** | **0.039** | 0.50 | 0.310 | 0.945 |
| C-CHVAE | **1.00** | 0.110 | 0.797 | **1.00** | 0.389 | 0.475 | 0.62 | 0.353 | 0.325 | **1.00** | 0.307 | **0.568** |
| VAE-CF | 0.50 | 0.115 | **0.586** | 0.38 | 0.356 | 0.460 | 0.39 | 0.202 | 0.293 | 0.34 | 0.310 | 0.577 |
| CounterNet | **1.00** | **0.075** | 0.725 | **1.00** | **0.257** | **0.354** | **1.00** | 0.121 | 0.259 | **1.00** | **0.222** | 0.626 |

## F  Second-order Evaluation

We define three additional *second-order* metrics which attempt to evaluate the usability of CF explanation techniques by human end-users. We posit that negligible feature differences (among continuous features) between instance $x$ and CF example $x'$ make it difficult for human end-users to use CF example $x'$ (as many of the recourse recommendations contained within $x'$ may not be actionable due to negligible differences). For example, human end-users may find it impossible to increase their *Daily_Sugar_Consumed* by 0.523 grams (if the value of *Daily_Sugar_Consumed* feature is 700 and 700.523 between $x$ and $x'$, respectively). As such, human end-users may be willing to ignore small feature differences between $x$ and $x'$.

To define our usability related metrics, we construct a user-friendly *second-order* CF example $x''$ by ignoring small feature differences (i.e., $|x_i - x'_i|$ is less than threshold $b$) between instance $x$ and CF example $x'$. Formally, let $x = \{x_1, x_2, .., x_d\}$ and $x' = \{x'_1, x'_2, .., x'_d\}$ be the features of the input instance and the CF example, respectively. Then, we use a threshold of $b$, and create a new data point $x'' = \{l_i = \mathbb{1}_{|x_i - x'_i| \leq b} x_i + \mathbb{1}_{|x_i - x'_i| > b} x'_i \, \forall i \in 1 \ldots d\}$, i.e., we replace all features $i \in \{1, d\}$ in CF example $x'$ with features in the original input instance $x$ for which $|x_i - x'_i| \leq b$. Our metrics for CF usability are defined in terms of $x$ and $x''$ as follows:

- *Second-Order Validity* is defined as the fraction of input instances on which $x''$ remains a valid CF example. High second-order validity is desirable, because it implies that despite ignoring small feature differences, the second-order CF example $x''$ remains valid.

- *Second-Order Proximity* is defined as the $L_1$ norm distance between $x$ and $x''$. It is desirable to maintain low second-order proximity because it indicates fewer cumulative modifications in the input space.

- *Second-Order Sparsity* is defined as the number of feature changes (i.e., $L_0$ norm) between $x$ and $x''$. High second-order sparsity enhances the interpretability of a CF explanation. Note that second-order sparsity is more important than the original sparsity metric, as the second-order CF example $x''$ ignores small feature changes in the continuous features, yielding fewer number of feature changes in the input space.

### F.1  Experimental Results

The evaluation of counterfactual usability measures the quality of the second-order CF example $x''$ which is created by ignoring negligible differences between input instance $x$ and the CF example $x'$. We use a fixed threshold $b = 2$ to derive the "sparse" second-order CF example $x''$, and compute the second-order evaluation metrics.

**Second-order validity.** Table 13 compares the second-order validity of CF examples generated by CounterNet and other baselines on all six datasets. Similar to results in Table 2, CounterNet performs consistently well across all six datasets on the validity metric, as CounterNet is the only CF explanation method which achieves over 93.7% second-order validity on all six datasets. In particular, CounterNet achieves ∼11% higher second-order validity than C-CHVAE (its closest competitor) on all six datasets. Further, CounterNet is the only CF method which achieves more than 90% second-order validity on the Breast Cancer dataset, whereas all post-hoc baselines perform poorly

Table 13: Evaluation of Usability of Counterfactual Explanations

| Datasets | Metrics | Methods | | | | | |
|---|---|---|---|---|---|---|---|
| | | VanillaCF | DiverseCF | ProtoCF | C-CHVAE | VAE-CF | CounterNet |
| **Adult** | **Validity** | 0.764 | 0.515 | 0.508 | **0.995** | 0.348 | **0.995** |
| | **Proximity** | **5.843** | 8.007 | 7.261 | 8.139 | 8.319 | 7.170 |
| | **Sparsity** | **4.445** | 5.297 | 5.181 | 5.771 | 5.869 | 5.148 |
| **HELOC** | **Validity** | **1.000** | 0.906 | **1.000** | 0.986 | **1.000** | 0.988 |
| | **Proximity** | 5.350 | 5.202 | 6.131 | 5.841 | 6.725 | **4.289** |
| | **Sparsity** | 20.304 | **9.979** | 18.514 | 18.166 | 20.546 | 17.020 |
| **OULAD** | **Validity** | **1.000** | 0.701 | 0.999 | 0.886 | 0.969 | 0.980 |
| | **Proximity** | 12.469 | 14.751 | 13.183 | 13.569 | 13.335 | **11.740** |
| | **Sparsity** | 23.618 | **17.516** | 23.360 | 24.696 | 18.162 | 22.472 |
| **Student** | **Validity** | 0.669 | 0.528 | 0.307 | **0.982** | 0.485 | **0.982** |
| | **Proximity** | **11.919** | 18.392 | 15.606 | 21.406 | 21.336 | 19.758 |
| | **Sparsity** | **6.840** | 9.313 | 7.896 | 10.847 | 10.951 | 10.043 |
| **Titanic** | **Validity** | 0.987 | 0.570 | 0.785 | **1.000** | 0.386 | 0.978 |
| | **Proximity** | 17.282 | 16.809 | 17.039 | 21.145 | 20.278 | **15.056** |
| | **Sparsity** | 9.906 | 9.632 | 9.960 | 12.359 | 11.964 | **9.215** |
| **Breast Cancer** | **Validity** | 0.699 | 0.196 | 0.329 | 0.615 | 0.210 | **0.937** |
| | **Proximity** | 1.313 | 0.890 | **0.655** | 3.618 | 2.089 | 1.422 |
| | **Sparsity** | 8.343 | **4.699** | 5.014 | 9.741 | 8.783 | 7.762 |

(none of them achieve second-order validity higher than 70%), despite the fact that three of these baselines (VanillaCF, DiverseCF, and ProtoCF) achieved more than 99% first-order validity on this dataset. This result demonstrates that CounterNet is much more robust against small perturbations in the continuous feature space.

**Second-order Sparsity and Proximity.** In terms of second-order sparsity, CounterNet outperforms two parametric CF explanation methods (C-CHVAE and VAE-CF), and maintains competitive performance against two non-parametric methods (VanillaCF and ProtoCF). Across all six datasets, CounterNet outperforms C-CHVAE and VAE-CF by ∼10% on the this metric. Moreover, the difference between the second-order sparsity achieved by CounterNet and VanillaCF (and ProtoCF) is close to 1%, which indicates that CounterNet achieves the same level of second-order sparsity as these two non-parametric methods. In terms of second-order proximity, CounterNet is highly proximal against baseline methods as it achieves the lowest proximity in HELOC, OULAD, and Titanic datasets (similar to results in Table 2).

**Cost-Invalidity Trade-off.** Figure 7 shows that Counter-Net positions on the bottom left of this figure, which illustrates that CounterNet can balance the cost-invalidity trade-off in the counterfactual usability evaluation. Notably, CounterNet outperforms all post-hoc methods in the second-order invalidity metric, and maintains the same level of second-order sparsity as VanillaCF and ProtoCF (∼1% difference). Moreover, although DiverseCF achieves ∼10% lower second-order sparsity value than CounterNet, it has ∼50% higher second-order invalidity than CounterNet. This results from DiverseCF's inability to balance the the trade-off between second-order invalidity and sparsity. This high second-order invalidity of DiverseCF hampers its usability, even though it generates more sparse explanations.

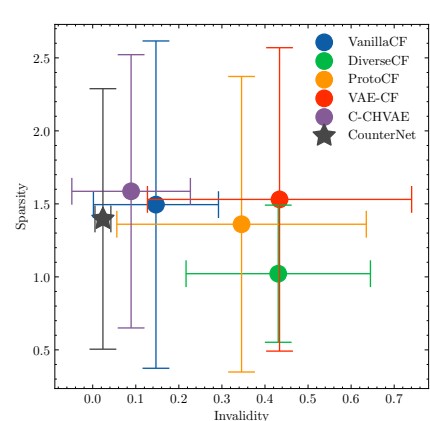

Figure 7: Illustration of trade-off between invalidity and sparsity across six datasets (methods at the bottom left are preferable).

# G   COUNTERNET
## UNDER THE MULTI-CLASS SETTINGS

In prior CF explanation literature, counterfactual explanations are primarily evaluated under the binary classification settings Mothilal et al. (2020); Mahajan et al. (2019); Upadhyay et al. (2021). However, it is worth-noting that CF explanation methods (including CounterNet) can be adapted to the multi-class classification settings. This section first describes the problem setting of the CF explanations when dealing with multi-class classification. Next, we describe how to train CounterNet for multi-class predictions and CF explanations. Finally, we present the evaluation set-up and show the simulation results.

## G.1   TRAINING COUNTERNET FOR MULTI-CLASS CLASSIFICATION

Given an input instance $x \in \mathbb{R}^d$, CounterNet aims to generate two outputs: (i) a prediction $\hat{y}_x \in \mathbb{R}^k$ for input instance $x$; and (ii) the CF example $x' \in \mathbb{R}^d$ as an explanation for input instance $x$. The prediction $\hat{y}_x \in \mathbb{R}^k$ is encoded as one-hot format as $\hat{y}_x \in \{0, 1\}^k$, where $\sum_i^k \hat{y}_x^{(i)} = 1$, $k$ denotes the number of classes. Moreover, we assume that there is a desired outcome $y'$ for every input instances $x$. Then, it is desirable that a CF explanation $y_{x'}$ needs to be predicted as the desired outcome $y'$ (i.e., $y_{x'} = y'$).

The objective for CounterNet in the multi-class setting remains the same as in the binary setting. Specifically, we expect CounterNet to achieve high *predictive accuracy*, *counterfactual validity* and *proximity*. As a result, we adjust loss functions from Eq. 1 as follows:

$$\mathcal{L}_1 = \frac{1}{N} \sum_{i=1}^{N} (y_i - \hat{y}_{x_i})^2$$
$$\mathcal{L}_2 = \frac{1}{N} \sum_{i=1}^{N} (\hat{y}_{x_i} - y')^2 \qquad (7)$$
$$\mathcal{L}_3 = \frac{1}{N} \sum_{i=1}^{N} (x_i - x_i')^2$$

Same as training CounterNet in the binary setting, we optimize the parameter $\theta$ of the overall network by solving the minimization problem in Eq. 1 to (except that we are switching to use loss functions in Eq. 7). Moreover, we adopt the same block-wise coordinate optimization procedure to solve this minimization problem by first updating for predictive accuracy $\theta' = \theta - \nabla_\theta(\lambda_1 \cdot \mathcal{L}_1)$, and then updating for CF explanation $\theta'' = \theta' - \nabla_\theta(\lambda_2 \cdot \mathcal{L}_2 + \lambda_3 \cdot \mathcal{L}_3)$.

## G.2   EXPERIMENTAL EVALUATION

**Dataset.** We use *Cover Type* dataset Blackard (1998) for evaluating the multi-class classification experiment. *Cover Type* dataset predicts forest cover type from cartographic variables. This dataset contains seven classes (e.g., Y=1, Y=2, ..., Y=7), with 10 continuous features. For CF explanation generation, we assume that cover type 5 (e.g., Y=5) is the desired class. The original dataset is highly imbalanced, so we equally sample data instances from each class.

**Results.** Table 14 compares the performance of counterNet and our two most competitive baselines (i.e., VanillaCF and C-CHVAE) in the evaluation for binary datasets (as found in Table 2 & 13). This table shows that CounterNet can achieve competitive performance against post-hoc CF explanation techniques in the multi-class classification settings. In terms of predictive accuracy, CounterNet performs comparably as the baseline methods with only ∼2% decrease (in average). In terms of validity and proximity, CounterNet can properly balance the cost-invalidity trade-off. Although CounterNet achieves higher proximity score than VanillaCF, it achieves 100% validity score. Compared to C-CHVAE, CounterNet achieves ∼80% lower proximity. Finally, CounterNet runs order-of-magnitudes faster than our two baseline methods. CounterNet runs more than 1000X and 3000X faster than C-CHVAE and VanillaCF, respectively.

Table 14: Results for CF explanation methods on Forester Cover Type dataset.

| Methods | Predictive Accuracy | Validity | Proximity | Running Time |
|---|---|---|---|---|
| VanillaCF | 0.911 | 0.921 | 0.379 | 1679.676 |
| C-CHVAE | 0.911 | 1.000 | 1.503 | 734.625 |
| CounterNet | 0.887 | 1.000 | 0.800 | 0.566 |

## H    IMPACT OF NEURAL NETWORK STRUCTURES

We further study the impact of the different neural network blocks. In our experiment, we primarily use multi-layer perception as it is a suitable baseline model for the tabular data. For comparison, We also implemented the CounterNet with Convolutional building blocks (i.e. replace the feed forward neural network with convolution layer). We implemented the convolutional CounterNet on the Adult dataset. To train the feed forward neural network with convolution layers, we set the learning rate as 0.03 and $\lambda_1 = 1.0$, $\lambda_2 = 0.4$, $\lambda_3 = 0.01$. The rest of the configuration is exactly the same as training CounterNet with MLP.

Table 15 shows comparison between CounterNet with convolutional building blocks (*CounterNet-Conv*) and multi-layer perceptions (*CounterNet-MLP*). The results indicate that CounterNet-Conv matches the performances of CounterNet-MLP. In fact, CounterNet-Conv performs slightly worse than CounterNet-MLP because convolutional block is not well-suitable for tabular datasets. Yet, CounterNet-Conv outperforms the rest of our post-hoc baselines in validity (with reasonably good proximity score). This illustrates CounterNet's potential real-world usage in various settings as it is agnostic to the network structures.

Table 15: Results for the CounterNet with Convolution layers on Adult dataset.

| Building Block | Predictive Accuracy | Validity | Proximity |
|---|---|---|---|
| CounterNet-Conv | 0.823 | 0.980 | 7.554 |
| CounterNet-MLP | 0.828 | 0.994 | 7.156 |

## I    COUNTERNET ON THE IMAGE DATASET

CounterNet is designed to generate counterfactual explanations for tabular datasets (the most common use case for CF explanations). We also experiment with CounterNet on the image datasets. This experiment uses the MNIST dataset: class "7" is used as the positive label, and class "1" is used as the negative label. Next, we apply the same CounterNet training procedure to generate image counterfactuals. Table 16 demonstrates the results of CounterNet on the MNIST dataset. CounterNet achieves 52.4% validity with 0.059 average $L_1$ distance. This result shows a current limitation of CounterNet as applying CounterNet as-is is ill-suited for generating image counterfactual explanations.

Table 16: CounterNet on the Image Datasets.

| | Validity | Proximity |
|---|---|---|
| **CounterNet** | 0.524 | 0.059 |

## J    REAL-WORLD USAGE.

We illustrate how CounterNet generates interpretable explanations for end-users. Figure 8 show an actual data point $x$ from the Adult dataset, and the corresponding CF explanation $x'$ generated by CounterNet. This figure shows that $x$ and $x'$ differ in three features. In addition, CounterNet generates $x''$ by ignoring feature changes that are less than threshold $b = 2$ (in practice, domain experts can help identify realistic values of $b$). Note that due to CounterNet's high second-order

validity, $x''$ also remains a valid CF example. After this post-processing step, $x$ and $x''$ differ in exactly two features, and the end-user is provided with the following natural-language explanation: "*If you want the ML model to predict that you will earn more than US$50K, change your education from **HS-Grad** to **Doctorate**, and reduce the number of hours of work/week from **48** to **33.5**.*"

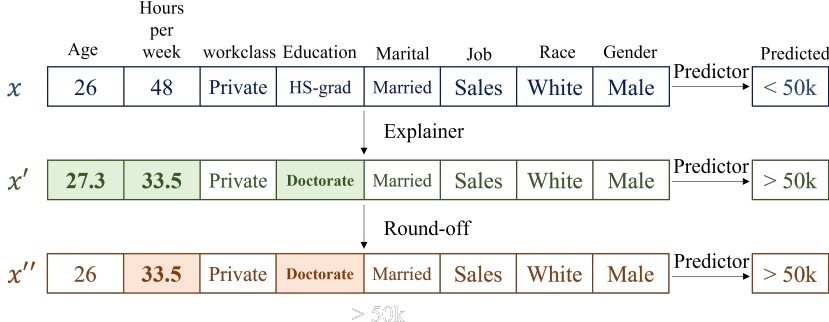

Figure 8: A counterfactual explanation from CounterNet.

