# OpenReview forum: "CounterNet: End-to-End Training of Prediction Aware Counterfactual Explanations"
_ICLR.cc/2023/Conference — Submitted to ICLR 2023_

### Official Review · Reviewer_Mece · 2022-10-23

**Confidence:** 5
**Correctness:** 1
**Technical Novelty And Significance:** 1
**Empirical Novelty And Significance:** 2
**Recommendation:** 3

**Clarity, Quality, Novelty And Reproducibility:**

The paper is well written and well organized. The problem it tackles is also interesting for the ML community.

One of my major concerns is the technical novelty, especially compared to to CounteRGAN [1], which is also highly efficient and uses a feed forward CF generator. This also begs the question why [1] is not included as a baseline to evaluate metrics such as Validity and Proximity.

In terms of runtime, note that methods like CounteRGAN [1] only need one single feedforward pass to generate the CE and therefore should be much faster than the proposed method. Unfortunately, [1] is not included as a baseline, making it difficult to reliably evaluate the efficiency of the proposed method compared to the state of the art.

This is also highly related to the authors’ claim that existing methods are slow because they rely on solving an optimization problem (one of the claimed major limitations). In fact [1] also contradicts the authors’ another mentioned limitation that existing methods are post-hoc and do not leverage training information. Note that [1] could also potentially be trained jointly with the predictor.

Lemma 3.1 is problematic or disconnected with Eq (3). Note that according to Eq (3) the prediction loss and the validity loss use different $y$’s as the target. Specifically, the prediction loss use $y_i$, the original label for $x_i$, as the target, while the validity loss use $\hat{y}_{x_i}$, the predicted label, as the target. However, in Lemma 3.1, $L_1$ and $L_2$ share the same target $y$.

Besides, the divergent gradient issue in Lemma 3.1 only happens when $x$ and $x’$ are close. However, we can almost be sure that $x$ and $x’$ are NOT close near the convergence point. Therefore, Lemma 3.1 is not related to the convergence point of the optimization problem at all. Rather, it is only related to the initialization point of the optimization problem.

For the second issue on adversarial examples, Lemma 3.2 is incomplete. Note that in practice, $L_1$ and $L_2$ are jointly optimized. In this case, the optimization process will try to train the predictor such that the prediction loss is minimized while training the CF generator to minimize the validity loss.

It would also be helpful to provide ablation study results on the performance when Eq (3) is optimized in the naïve way. It seems this is included as CounterNet-SingleBP in Table 4.

Table 1 shows that CounterNet has a disadvantage in terms of predictive accuracy when compared to post-hoc approaches, since post-hoc approaches could preserve the predictive accuracy of the base models fully.

According to Table 2, it seems CounterNet does not have consistent improvement upon baseline methods.

The writing is inconsistent. In the abstract, the authors mentioned that existing CF methods suffer from two major limitations. In the introduction section, the number of major limitations becomes three.

The authors argue that post-hoc explanations are a major limitation in that they do not full make use of the information during training of the ML model. I find it somewhat vague and unconvincing.

[1] CounteRGAN: Generating Counterfactuals for Real-Time Recourse and Interpretability using Residual GANs. UAI 2022.


**Strength And Weaknesses:**

Strength

+ Generating counterfactual explanations is an important and interesting problem in the ML community.

+ The design of the model with three terms, prediction loss, validity loss, and change loss, makes sense.

Weaknesses

- The proposed method has limited technical merit, especially compared to recent methods such as CounteRGAN [1].

- Some claims in the paper on major limitations of existing methods are not true.

- Important and highly relevant baselines are missing.

- Lemma 3.1 is incorrect or irrelevant to Eq (1) (see below for details).


**Summary Of The Paper:**

This paper tackles the problem is generating counterfactual (CF) explanations for neural networks. The authors propose a model that consists of an encoder, a predictor, and a CF generator. Correspondingly these models are trained by optimizing an objective function with three loss terms, prediction loss, validity loss, and change loss. Experiments show that CounterNet can generate CF explanations with reasonable performance.

**Summary Of The Review:**

This paper tackles the problem is generating counterfactual (CF) explanations for neural networks. This is an interesting and important problem in the ML community. Overall the model design makes sense to me. My major concerns include the absence of highly related state-of-the-art baselines, several false claims or over-claims in the abstract and introduction, and the limited technical merit, especially given a related work CounteRGAN.

---

> ### Author Response · Authors · 2022-11-11
> **Response to Reviewer Mece**
>
> Thank you for your detailed and insightful feedback. Below, we provide a question-by-question rebuttal to your concerns.
>
> > Comparison to CounteRGAN
> >
>
> Thank you for bringing CounteRGAN to our attention. CounteRGAN is yet another *post-hoc parametric* method (similar to *VAE-CF*, *C-CHVAE*, and *VCNet* as compared in the initial submission), and it **does not** leverage training information (unlike CounterNet). Nevertheless, based on your suggestion, we have implemented and compared CounteRGAN as a baseline in our paper (please see the updated paper version). In short, CounteRGAN achieves ~45% poorer validity and ~76% poorer proximity as compared to CounterNet (new Table 2), and achieves a significantly poorer cost-invalidity trade-off (new Figure 3). In addition, we found that CounteRGAN is more than ~3.4X slower (in runtime) than CounterNet (see new Table 3).
>
> > On Lemma 3.1
> >
>
> First, we want to point out that Lemma 3.1 highlights the challenges experienced **during the training** of CounterNet (instead of its status in near convergences). Thus, it may be true that $x$ and $x’$ are not very very close to each other at convergence, $x$ and $x’$ can be very close to each other since we explicitly optimize for proximity (i.e., $\mathcal{L}_3$). This theoretical analysis is confirmed by our ablation analysis (in Table 4) where *CounterNet-SingleBP* (optimize for Eq.3 directly) **performs poorly in comparison*.*
>
> Second, as per your suggestion, we weaken the assumption on $\hat{y}_x$ as now $\mathcal{L}_1$ and $\mathcal{L}_2$ have the same loss formulation in Eq. 3. Instead of assuming $y = \hat{y}_x$, now we can prove that the original statement holds true as long as the model makes the correct prediction (i.e., $|\hat{y}_x-y| < 0.5$), which is feasible during the training as minimizing $\mathcal{L}_1$ ensures $y$ and $\hat{y}_x$ close.
>
> > On Lemma 3.2
> >
>
> $\mathcal{L}_1$ and $\mathcal{L}_2$ are not optimized as part of the same gradient update. Instead, $\mathcal{L}_1$ is optimized in the first backward pass, and $\mathcal{L}_2+\mathcal{L}_3$ is optimized in the second backward pass. Therefore, we show in Lemma 3.2 how optimizing $\mathcal{L}_2$ (under the assumption that $x\to x’$, which is equivalent to optimizing $\mathcal{L}_3$ to near 0) w.r.t. $\theta$ degrades the adversarial robustness of the network.
>
> > Ablation study results on the performance when Eq (3) is optimized in the naïve way
> >
>
> As you rightly mention, we have already included this ablation as *CounterNet-SingleBP* in Table 4.
>
> > CounterNet has a disadvantage in terms of predictive accuracy
> >
>
> We would like to emphasize that the percentage difference between CounterNet and post-hoc approaches (in terms of predictive accuracy) is only ~0.5% on average, which is a minimal difference. CounterNet also achieves a higher accuracy (than the base model of post-hoc approaches) on the Credit dataset (see Table 1).
>
> > CounterNet does not have consistent improvement upon baseline methods.
> >
>
> We respectfully disagree. CounterNet is optimized for validity and proximity. And it achieves the **highest** possible validity of 100% on **all 4 datasets** (Table 2). In terms of Proximity, CounterNet achieves the **best** proximity scores on **3/4 of datasets**. Overall, CounterNet is the best performing method in balancing the cost-invalidity trade-off (Figure 3). This shows that CounterNet outperforms all baselines by generating CF examples with the highest validity and best proximity scores.
>
> > Unconvinced about post-hoc limitation
> >
>
> Please refer to our answer to Reviewer [9tKf](https://openreview.net/forum?id=PocqkbIelt&noteId=kGsQCBJ_cL) about our clarification of why post-hoc methods do not work as well as CounterNet.
>
> To summarize our argument, post-hoc methods **cannot** make use of decision boundary information during the training of the ML model (because the model is pre-trained and black-box, by the post-hoc assumption). Intuitively, it is impossible for any CF generation method to find the theoretically optimal CF example $x’$ (for input x) without making use of this decision boundary information (OR without having the ability to modify the decision boundary). As a consequence, post-hoc methods also **cannot** find the optimal $x’$ as a CF example.

---

> ### Author Response · Authors · 2022-11-18
> **Look forward to your post-rebuttal discussion**
>
> Dear Reviewer Mece,
>
> Thank you very much for your thoughtful feedbacks. We hope our responses addressed your concerns adequately. If you have additional comments/questions, feel free to let us know. We will try our best effort to address them.

---

> ### Comment · Reviewer_Mece · 2022-12-11
> **Thank you for your response**
>
> I have read the author response and other reviews. I would like to thank the authors for their response, which is helpful in clarifying some of the confusions.
>
> However, my concerns on the correctness of Lemma 3.1 and the CounteRGAN baseline is still not very well addressed. For example, the author provide very limited results from CounteRGAN (which is understandable due to time constraint), and from the results it is a bit counterintuitive that CounteRGAN is much slow at inference time, given it is actually a simply method. I also with some of the comments from fellow reviewers, e.g., related work from Schleich et al.
>
> I therefore would like to keep my score unchanged.

---

### Official Review · Reviewer_MUfa · 2022-10-24

**Confidence:** 4
**Correctness:** 3
**Technical Novelty And Significance:** 3
**Empirical Novelty And Significance:** 3
**Recommendation:** 10

**Clarity, Quality, Novelty And Reproducibility:**

The work is presented extremely clearly and, to the best of my knowledge, represents a clearly novel method for the field. Reproducibility should be high, given the publication of the code (although not personally tested). I consider the quality of this work to be very high.

**Strength And Weaknesses:**

# Strengths:

1. Clear and concise explanation of the Counterfactual Explainability setting.

2.  The key limitations of previous methods are clearly presented, as well as how the new method is able to overcome them. The limitations of this new method are also explicitly stated.

3. Much appreciated is the theoretical analysis of Convergence and Adversarial Examples, which gives depth to the paper.

4. Experiments seems complete, with multiple datasets, many baselines and ablation analysis about design choices.

# Weaknesses:

1. Although some are well shown, I think some limitations of the model need to be made more explicit. W1a) First of all, this method is not suitable for already trained networks, which makes it useless if you cannot/will not replace the model you have with this one.
     * Does the training run slower than that of a standard NN? If so, this should be highlighted as a (small) limitation. The time of just training the predictor could be added to Table 3.
     * I also think it should be made explicit that, considering the manifold, VCNet is still very competitive, also managing to have perfect validity, but lesser manifold on 3 of the 4 datasets.
     * While briefly stated in a footnote, there is no experimental proof that this architecture works well in domains where alternatives neuronal structures performs best. The authors try their method with a Convolutional block, but with tabular datasets; as they say "convolutional block is not well-suitable for tabular datasets”. Likewise, is not explicit if the CounterNet used for the Image dataset is a convolutional one (Appendix I). Since this is not a good enough experimental proof to state that the method works well (as CF Generator) with other types of neural networks (as you say for images), it is necessary in my opinion to make this limitation explicit in the main paper and not only in the supplementary.

2. “Note that passing px through the CF generator network is analogous to feeding information about the decision boundary to the CF example generation procedure, which leverages this knowledge to find high-quality CF examples x′” The reason why px contains boundary information is not clear. Likewise, is not theoretically explained why the CF generator needs both zx and px (apart from the experiments in the Ablation Analysis section).

3. Some claims in the Experimental evaluation section should be sustained via statistical tests.

**Summary Of The Paper:**

The work proposes a new method for generating Countefactuals for neural networks. The method consists of designing the network by having a part dedicated to the creation of counterfactuals, which is trained together with the predictive part. Through reasoned design and a special training, this new network, called CounterNet, is able to achieve accuracy on a par with other models. It can also generate counterfactuals in much less time than current post-hoc methods and achieve competitive values of widely used metrics from prior literature.

**Summary Of The Review:**

The strengths of the work far outweigh the weaknesses I have identified. Regarding these, they are mostly on the clarity of certain images or statements, which I strongly encourage the authors to follow, to make the work even more polished than it is now.

## Minor remarks:
1. In Introduction, I would advise some minor specifications:
    * It should be specified that x2’ is an instance with different (or opposite) label.
    * “More generally, CF examples with low (high) cost of change imply high (low) invalidities”. While usually true, this is not always the case. I would rephrase it, including “often”, “usually” or a synonym.
    * “Next, we describe key limitations of prior CF techniques.” Seems redundant/ a repetition.
    * “First, all prior methods” I’ll add, just once and here, “to our best knowledge” or similar.
    * “Post-hoc explanation techniques are agnostic” Not every method is agnostic.
    * “Consequently, such a post-hoc procedure does not properly balance the cost-invalidity trade-off (as explained above), causing shortcomings in the quality of the generated CF explanations” This would need a citation

2. In Section 3:
    * “output is binary-valued, then yˆx and yˆx′ should take on opposite values (i.e., yˆx + yˆx′ = 1)”. Since in a binary classification task, labels can also be (+1) and (-1), I would write “output is binary-valued (0, 1)” or similar
    * While very well designed, Figure 2 hides two flaws. First, it does not include a yx’ used just for training, that is indeed cited in the text; probably adding a dotted line at the end of Predictor will fix this. Second, it should be highlighted that the connection from the Predictor to the CF Generator happens only once; otherwise, the network, in training regime, will continuously produce Counterfactuals of its own Counterfactuals in loop. Even in text, the “feedback loop” from x’ to input is mentioned, but not the px to CF Generator.
    * “generate a latent vector z ∈ Rk (s.t. k < d)”, “and up-samples” While generally true, there’s no need to specify k < d, since, to my understanding, nothing prevents the network to work even if k ≥ d, unless you have experiments that prove otherwise.
    * I think that three hyper-parameters are not actually needed for the loss (1), cause what’s important is the relative weight of the three losses. The additional parameter probably just ensure that there are multiple optimal configuration, fixing other hyperparameters (e.g. learning rate, decay rate, etc.). However, I don't think it should be changed.
    * “Proof of Lemma 3.1 and 3.2 can be found in Appendix A.” It’s strange to mention Lemma 3.2 in the Lemma 3.1 section. I would move this sentence to the previous introductory paragraph, that of subsection 3.3
    * “we update network weights θ = {θh, θf , θg} twice” This is in contrast with the sentence “we only update the weights in the CF generator θg”, cause you seem to update twice just the θg weights. But, if looking carefully at L1 loss, it doesn’t depend on θg, so each weight is actually updated just once, if you freeze the gradient updates.

3. Experimental evaluation
    * The Ablation Analysis section is really appreciated, but in my opinion needs slightly more clarity. Perhaps even just starting a new line every time a modification is mentioned would help.
    * "reaches the theoretical upper bound of robustness (i.e., the robustness of base model)". This statement is incorrect for me. First of all, I do not understand why the upper bound of robustness should be the robustness of base model. Even if it were, CounterNet exceeds it, so it cannot be an upper bound. Secondly, there is no theoretical study of the bounds of robustness, so I would avoid such a mathematically specific statement.

4. Appendix:
    * In Section G, “methods with only 2 ̃% decrease (in average).” The tilde ~ should be before the 2%

---

> ### Author Response · Authors · 2022-11-11
> **Response to Reviewer MUfa**
>
> We thank reviewers for your careful read and detailed feedback on our paper, and thank you for the encouraging assessment. We have incorporated all your suggestions into the paper. Please see the answers to individual points below.
>
> > This method is not suitable for already trained networks.
>
> We agree that CounterNet and post-hoc methods have somewhat different motivations:
>
> *While post-hoc methods are designed for generating CF explanations for trained black-box ML models (whose training data and model weights might not be available), CounterNet is most suitable when the ML model developers aspire to build prediction and explanation module from scratch, where the training data can be exploited to optimize the generation of CF examples.*
>
> We have added the above line to Section 5.
>
> > Training time of CounterNet
>
> CounterNet’s training time comparison is shown in Table 9 of the Appendix, where we show that it would take approximately $211$ seconds (<4 minutes) to train CounterNet on the Adult dataset (which is our largest dataset). In comparison, the base predictive model (which does not account for multiple objectives) has a training time of 71 seconds (see Table 9 for further results).
>
> > Highlight VCNet’s Competitiveness
>
> Thank you for your suggestion. We have added a line to this effect in Section 4.1.
>
> > CounterNet’s Performance with Alternate Neuronal Structures
>
> We agree, even though Appendix H uses a convolution-based CounterNet, it is not strong experimental proof, and requires future efforts to effectively train these models. We have added a line to this effect in the footnote.
>
> > Reason why px contains boundary information is not clear
>
> $p_x$is the dense representation of the predictor network, which implicitly conveys information about the decision boundary of the predictor network. We have made this clearer in the paper.
>
> > Reason why zx and px are both needed for CF generator
>
> Intuitively, to generate good counterfactuals, we need two pieces of information (i) an input example $x$ (whose representations are stored in $z_x$), and (ii) the predictive network for generating the CF explanations (whose representations are stored in $p_x$). Thus, in the CF generator, both these things are passed to optimize the CF generator. We have made this clearer in the paper.
>
> > Some minor remarks
>
> We appreciate your very careful read of the paper, and have amended the paper based on your suggestions.

---

> ### Author Response · Authors · 2022-11-18
> **Look forward to your post-rebuttal discussion**
>
> Dear Reviewer MUfa,
>
> Thank you very much for your thoughtful feedback. We hope our responses addressed your concerns adequately. If you have additional comments/questions, feel free to let us know. We will try our best effort to address them.

---

### Official Review · Reviewer_9tKf · 2022-10-24

**Confidence:** 4
**Correctness:** 2
**Technical Novelty And Significance:** 2
**Empirical Novelty And Significance:** 2
**Recommendation:** 3

**Clarity, Quality, Novelty And Reproducibility:**

The writing is clear and good. The reproducibility statement is also detailed. However, the novelty is limited.

**Strength And Weaknesses:**

Strength:
 - It proposes a new learning framework to generate counterfactual explanations(CF) for ML models, which facilitates service providers to specialize CF explanation techniques that can leverage the knowledge of their particular ML model.
- They theoretically analyze the joint objective and address two challenges in the jointly end-to-end training procedure. It may offer some insights into the reasonability of the iterative training scheme when jointly training two modules with different roles.

Weaknesses
- On the method:
    - The idea of intrinsic explainability is not new, though may have not been applied in generating CF explanations directly.
    - The so-called “block-wise coordinate descent procedure” is indeed an iterative training scheme, which is very common when handling two or more modules.
    - While the paper emphasizes the proposed framework can achieve better cost-invalidity trade-off than the post-hot methods, it’s not clear why the framework can achieve that. The paper says “Second, in the post-hoc CF explanation paradigm, the optimization procedure that finds CF explanations is completely uninformed by the ML model training procedure (and the resulting decision boundary). Consequently, such a post-hoc procedure does not properly balance the cost-invalidity trade-off”, but it seems impossible and infeasible to obtain the decision boundary of the black-box model.
- Others:
    - A very similar paper seems to have been published on ICML2021.
    - GitHub link provided in the paper seems to leak information about the author.
    - There is a typo in the header of Table 5.


**Summary Of The Paper:**

This paper focuses on generating counterfactual explanations for model predictions, and then proposes an end-to-end learning framework that learns an explanation generator while training the predictive model. They also analyze the challenges in jointly training the predictor and the explanation generator and propose an iterative training procedure.

**Summary Of The Review:**

As the novelty is limited and the main argument(on cost-invalidity trade-off) is not well clarified, I am lean on the negative side.

---

> ### Author Response · Authors · 2022-11-11
> **Response to Reviewer 9tKf**
>
> We thank the reviewers for your detailed and insightful feedback. Below, we provide a question-by-question rebuttal to your concerns.
>
> > 1. On the novelty of our approach
>
> We respectfully disagree about the lack of novelty in the paper. Please see answers to your main questions below.
>
> **“Idea of Self-Explainable (or Intrinsic) Explainability is not new.”**
>
> You are right in pointing this out that self-explainable models are not new ideas. However, as you rightly mention, the idea of intrinsic interpretability has never been applied before to generating CF explanations. As mentioned in our paper, post-hoc approaches represent the dominant paradigm for generating CF explanations, whereas many real-world use cases favor self-explainable interpretability, e.g., service providers wanting to become compliant with the “Right to Explanation” in EU-GDPR, etc. As a departure from this prevalent post-hoc paradigm, our work makes the first-ever novel attempt at achieving self-explainable interpretability in CF explanations (which is a highly non-trivial task).
>
> “’**block-wise coordinate descent procedure’ is very common”**
>
> Again, we agree that similar ideas of block-wise coordinate descent have been used in the past. However, it is highly non-trivial to design a specific block-wise coordinate descent strategy that is well suited to a completely new domain.
>
> The main technical novelty in our work lies in the theoretical analysis of CounterNet’s loss function, which uncovers the divergent gradient problem and its susceptibility to adversarial examples. We design a novel block-wise coordinate descent strategy that is consistent with the theoretical insights gleaned from Lemmas 3.1 and 3.2. Further, in our ablation analysis (see Table 4), we show that our theoretically motivated coordinate descent strategy also performs the best empirically.
>
> In addition, we propose a novel neural network architecture that enables (for the first time) the joint training of predictions and CF explanations. Finally, we also conduct a rigorous experimental evaluation of a wide variety of baseline methods that shows that CounterNet achieves SOTA performance on several popular evaluation metrics.
>
> > 2. Why CounterNet enables a better cost-invalidity trade-off?
>
> The intuition for why CounterNet balances the cost (or proximity)-invalidity trade-off is shown in Figure 1. As shown in this figure, the optimal CF example for input data point x (red color) would be a point x’ (which is just minimally across the decision boundary from x) since it would have the best possible proximity to x. Intuitively, it is impossible to find this optimal x’ without knowing what this decision boundary is (or without having the ability to modify the decision boundary).
>
> As you rightly mention, post-hoc approaches cannot access the decision boundary of the ML model (because it is a black-box model, by assumption). Therefore, this optimal x’ can be more challenging than CounterNet (as we shown in the empirical results).
>
> ## Minor Remarks
>
> We have fixed the typo in Table 5. We delete the GitHub repository that potentially leaks the author’s information, and reuploads the code as the supplement materials.

---

> ### Author Response · Authors · 2022-11-18
> **Look forward to your post-rebuttal discussion**
>
> Dear Reviewer 9tKf,
>
> Thank you very much for your thoughtful feedback. We hope our responses addressed your concerns adequately. If you have additional comments/questions, feel free to let us know. We will try our best effort to address them.

---

### Official Review · Reviewer_Ycp6 · 2022-11-03

**Confidence:** 4
**Correctness:** 3
**Technical Novelty And Significance:** 2
**Empirical Novelty And Significance:** 2
**Recommendation:** 3

**Clarity, Quality, Novelty And Reproducibility:**

Clarity: The paper is mostly easy to follow.

Quality: I think the quality in terms of explaining the added value and the design choices can be significantly improved. See the detailed comments and suggestions above.

Novelty: The idea of generating explanations as a part of the forward pass has been around for some time. Though I do not recall it being used in the context of CFs. Nonetheless, this seems to be a rather minor novelty.

Reproducibility: The paper links to the open source code though I have not checked it.

**Strength And Weaknesses:**

### Strengths

1. The topic of the paper (generating CFs) is important and timely.

### Weaknesses

Overall, the paper makes a bunch of adhoc choices when setting up the architecture and the loss function. The added value over the other methods is not very clear. Also, the proposed architecture does not support important criteria like diversity. See detailed comments below:

1. First, it is not very clear if improving the runtime is such an important aspect that requires training the model from scratch. From Table 3, it looks like Vanilla CF seems to be able to generate a CF within 1-2 seconds which seems quite reasonable. Also, adding the CF generation to model training seems to prolong the training time (due to multiple objective and alternative updates). How large was this additional training time?

2. Some of the choices could be justified better. For instance, why consider the non-standard squared loss in the training objective. The paper provides some ablation analysis but the reasoning behind moving from the standard cross entropy loss is not very clear. Similarly, why should the change loss be the squared loss (is it the L2 norm)? Why not consider the sparsity inducing distance like in Wachter et al? Similarly, should we expect the proposed method to be sparse in Section 4.1 when the objective function contains no sparsity term itself?

3. Section 3.3: Why is adversarial robustness is a special concern here? Aren’t usual NNs also prone to adversarial perturbations? I see the point that the addition of the validity loss might exacerbate the robustness. But then the paper should quantify the additional lack of robustness.

4. Section 3.3: How does the alternate update strategy solve the divergent gradient problem? Since the models two objectives are in conflict with each other, is there an equilibrium point? How do you detect convergence?

5. When measuring the manifold distance, it is not clear why the L1 metric was chosen. Why not L2 distance? Why only consider the 1-NN?

6. Recent work on CFs seems to support more criteria than validity and proximity. For instance, Mothilal et al consider diversity. Schleich et al (https://arxiv.org/pdf/2101.01292.pdf) consider a combination of distance function to cap the number of feature changes and the max. change in the feature. Can these criteria be included in CounterNet?

7. Since many of the gradient based methods highly depend on the init point (due to non-convexity of the CF objective), simply restarting with different random seeds could help with validity. It would be greatly helpful to add the restarting strategy to see how much the validity improves.

8. Different CF methods come with different hyperparameters (e.g., balancing between sparsity and validity). How were these HPs selected?

**Summary Of The Paper:**

The goal of the paper is the provide counterfactual explanations (CFs) for neural classification models. The paper makes the point that existing methods for generating CFs are posthoc -- so they are not aware of the decision boundary of the model. As a result, the approximations taken by these models lead to subpar CFs. The posthoc search also means that the process of generating CFs is time consuming. To overcome these issues, the paper proposes to generate CFs as a part of the model forward pass. Specifically, the model architecture and the objective is modified to generate CFs. Experiments are shown comparing the performance with several existing methods.

**Summary Of The Review:**

Overall, it is not clear if the problem solved in the paper (improving runtime and validity of CFs) is a relevant problem to begin with. The paper also could be improved in terms of conveying the design choices which seem quite adhoc. For these reasons, I don't think the paper is ready yet.

---

> ### Author Response · Authors · 2022-11-11
> **Part II of Response to Reviewer Ycp6**
>
> > 3. Why is adversarial robustness a special concern here?
>
> You are right in pointing out that the addition of the validity loss exacerbates the adversarial robustness issue. But we do already quantify the additional lack of robustness (caused by the validity loss) in Figure 4 of the paper. This figure shows that the difference in perturbation stability between CounterNet (which includes validity loss) and the base predictive model (which DOES NOT include validity loss) is less than 0.5%. Further, CounterNet-NoFreeze (a variant of CounterNet in which we do not freeze gradient updates in encoder and predictor networks during our second backward pass) achieves 30.9% lower perturbation stability than the Base Predictive Model. Thus, due to the addition of the validity loss, CounterNet experiences a minimal (~0.5%) additional lack of robustness as compared to the Base Predictive model.
>
> > 4. How does the alternate update strategy solve the divergent gradient problem?
>
> As shown in Lemma 3.1, if we calculate cumulative gradient across L1, L2, and L3, it suffers from a vanishing gradient problem, because L1 and L2’s gradients are divergent. To fix this, our alternate update strategy separates the gradient calculation of L1 (which is done in the first backward pass) from the gradient calculation of L2, L3 (which is done during the second backward pass). This ensures that we don’t do gradient updates with diminishing/vanishing gradients.
>
> > 5. why the L1 metric was chosen for measuring the manifold distance?
>
> We chose the L1 metric to remain consistent with choices made in prior work (as mentioned in the main paper). Nevertheless, we will shortly be updating the paper to report the performance of CounterNet and baselines on manifold distance computed with L2 metric and K-NN (with K=5).
>
> > 6. Recent work on CFs seems to support more criteria than validity and proximity (e.g., diversity).
>
> We consider these desirable aspects as future extensions to our paper, as we stated in Section 5. In theory, we can integrate these aspects by adding additional loss functions to our current formulation (e.g., `dpp_diversity` loss for diversity).
>
> > 7. Restarting for gradient-based methods.
>
> Thank you for this suggestion. We are running experiments based on this suggestion currently and will update our rebuttal and paper accordingly with new experimental results.
>
> > 8. How were these hyperparameters selected?
>
> Hyperparameters are selected via grid search. Details on the grid search are already mentioned in the Appendix.

---

> > ### Author Response · Authors · 2022-11-18
> > **Additional Experiments**
> >
> > Thank you again for your valuable feedback on the paper. Here, we provide additional experiments as per your suggestions.
> > > Restarting for gradient-based methods.
> > >
> >
> > Thank you again for your suggestion. Here, we report applying the random restart strategy for VanillaCF (namely, *VanillaCF-Restart*). We randomly initiate five times and select the best-generated counterfactual example (i.e., the CF example with the lowest object loss).
> >
> > In general, *VanillaCF-Restart* underperforms as compared to CounterNet in balancing the cost-invalidity trade-off.
> >
> > ### Adult
> >
> > |  | Validity | Proximity | Sparsity | Manifold |
> > | --- | --- | --- | --- | --- |
> > | VanillaCF | 0.76 | 0.202 | **0.556** | **0.57** |
> > | VanillaCF-Restart | 0.86 | 0.269 | 0.660 | 0.98 |
> > | CounterNet | **1.00** | **0.196** | 0.644 | 0.64 |
> >
> > ### Credit
> >
> > |  | Validity | Proximity | Sparsity | Manifold |
> > | --- | --- | --- | --- | --- |
> > | VanillaCF | 0.92 | **0.123** | 0.841 | 0.59 |
> > | VanillaCF-Restart | 0.91 | 0.181 | **0.745** | 1.48 |
> > | CounterNet | **1.00** | 0.132 | 0.912 | **0.56** |
> >
> > ### HELOC
> >
> > |  | Validity | Proximity | Sparsity | Manifold |
> > | --- | --- | --- | --- | --- |
> > | VanillaCF | **1.00** | 0.154 | 0.883 | 0.71 |
> > | VanillaCF-Restart | 0.946 | 0.189 | 0.853 | 1.43 |
> > | CounterNet | **1.00** | **0.125** | **0.740** | **0.56** |
> >
> > ### OULAD
> >
> > |  | Validity | Proximity | Sparsity | Manifold |
> > | --- | --- | --- | --- | --- |
> > | VanillaCF | 1.00 | 0.101 | 0.762 | 1.30 |
> > | VanillaCF-Restart | 0.894 | **0.050** | **0.587** | 1.72 |
> > | CounterNet | **1.00** | 0.75 | 0.725 | **0.87** |
> >
> > > Manifold distance.
> > >
> >
> > As mentioned in the previous rebuttal, we choose $L_1$ distance and 1-NN for consistency with prior work. Nevertheless, we compare baseline methods and CounterNet on $L_1$/$L_2$ distance and 1-NN/5-NN on the adult dataset. Please refer to the table below.
> >
> > |  | Manifold ($L_1$,1-NN) | Manifold ($L_1$,1-NN) | Manifold ($L_2$,5-NN) | Manifold ($L_2$,5-NN) |
> > | --- | --- | --- | --- | --- |
> > | VanillaCF | 1.438 | 2.261 | 0.927 | 1.283 |
> > | DiverseCF | 2.300 | 2.625 | 1.319 | 1.449 |
> > | ProtoCF | 0.920 | 1.256 | 0.656 | 0.843 |
> > | C-CHVAE | 1.395 | 1.797 | 0.925 | 1.137 |
> > | VAE-CF | 1.614 | 2.059 | 1.034 | 1.251 |
> > | CounterNet | 0.928 | 1.676 | 0.711 | 1.060 |

---

> ### Author Response · Authors · 2022-11-11
> **Part I of Response to Reviewer Ycp6**
>
> Thank you for your detailed and insightful feedback. Below, we provide a question-by-question rebuttal to your concerns.
>
> > 1 (a) Runtime is not such an important aspect to require training model from scratch…
>
> First, we clarify that the potential improvement of runtimes is not the only reason behind training the model from scratch. Doing so enables us to achieve (i) better validities, (ii) better proximities, (iii) faster runtimes (and we achieve all of this without sacrificing predictive accuracy, as shown in Section 4.1).
>
> Nevertheless, we argue that runtime is the absolute most crucial factor when algorithmic recourse applications are deployed in end-user-facing devices such as smartphones, tablets, etc. Deploying recourse applications and CF explanations systems in end-user-facing devices is reasonable since the intended recipient of recourse recommendations are average end-users who are negatively impacted by algorithmic decisions.
>
> In fact, there is a plethora of prior research to support our argument: (i) Google estimates that an additional 500ms delay per user transaction results in up to 20% loss of traffic [1]; (ii) Amazon estimates that every 100ms delay on their smartphone and web browser applications would cause 1% annual sales loss, which amounts to 745 Million dollars per year [2]; (iii) Packet Zoom estimated the cost of 1 additional second of latency in mobile applications at 0.08 dollar per user [3].
>
> We have updated the paper with these additional references to address your concern.
>
> > 1 (b) Clarification on Training Time of CounterNet
>
> We have already provided model training time numbers in Table 9 of the Appendix, where we show that it would take approximately $211$ seconds (<4 minutes) to train CounterNet on the Adult dataset (which is our largest dataset). In comparison, the base predictive model (which does not account for multiple objectives) has a training time of 71 seconds (see Table 9 for further results). Nevertheless, a training time of 211 seconds is quite reasonable since training time is a one-time up-front cost; after the model is trained, test-time inference happens very fast (as shown in Table 3).
>
> > 2 (a) Justification/Intuition for using MSE as compared to cross-entropy loss?
>
> In addition to our ablation analysis (which empirically shows that MSE performs better than cross-entropy), there is also a lot of prior research which shows that MSE brings about significant improvements on a wide variety of non-vision tasks, as compared to cross-entropy (CE) loss. In addition, MSE loss seems to be less sensitive to randomness in initialization, and also more robust to noise (as compared to CE)[4, 5]. Also, CE is more prone to overfitting [6].
>
> > 2 (b) Choice of L2 loss vs sparsity loss for Change Loss Term
>
> In choosing our change loss term, we wanted to capture the notion of the “cost of change” required to modify input x to CF example x’. Both L2 loss and sparsity loss capture this notion of “cost of change”. In fact, L2 loss and sparsity loss are highly correlated, so models optimized for L2 loss also achieve good sparsity (and vice versa). Finally, L2 loss was chosen over the sparsity loss (to be included in the main paper) simply because it represents a smoother objective (which is easier to train on), and our empirical analysis confirms this fact.
>
> To further address your concern, we supplement our ablation analysis by considering CounterNet’s performance under a wide variety of possible loss function choices (see Table 11 in the Appendix).
>
> ## Reference
>
> [1] Zhao, Y., Laser, M. S., Lyu, Y., & Medvidovic, N. (2018, May). Leveraging program analysis to reduce user-perceived latency in mobile applications. In *Proceedings of the 40th International Conference on Software Engineering* (pp. 176-186).
>
> [2] Arapakis, I., Park, S., & Pielot, M. (2021, March). Impact of Response Latency on User Behaviour in Mobile Web Search. In *Proceedings of the 2021 Conference on Human Information Interaction and Retrieval* (pp. 279-283).
>
> [3] [https://medium.com/codavel-blog/mobile-app-performance-1-second-delay-costs-0-08-per-user-74bf12f49eec](https://medium.com/codavel-blog/mobile-app-performance-1-second-delay-costs-0-08-per-user-74bf12f49eec)
>
> [4] Hui, L., & Belkin, M. (2021). Evaluation of neural architectures trained with square loss vs cross-entropy in classification tasks. In Proceedings of the International Conference on Learning Representations (ICLR), 2021.
>
> [5] Ghosh, A., Kumar, H., & Sastry, P. S. (2017, February). Robust loss functions under label noise for deep neural networks. In *Proceedings of the AAAI conference on artificial intelligence*
>  (Vol. 31, No. 1).
>
> [6] Baena, R., Drumetz, L., & Gripon, V. (2022). Preserving Fine-Grain Feature Information in Classification via Entropic Regularization. *arXiv preprint arXiv:2208.03684.*

---

### Author Response · Authors · 2022-11-18
**Rebuttal Summary**

We appreciate valuable and insightful feedback from reviewers. We have significantly updated our paper based on suggestions by the reviewers, and have added several additional experiments to the updated paper and Appendix (as suggested reviewers). Reviewers can view the revision history on this [page](https://openreview.net/revisions?id=PocqkbIelt) or check the exact changes via this [link](https://openreview.net/revisions/compare?id=PocqkbIelt&left=YxtUX5BC6&right=L-t7kF60v&pdf=true).

Here we provide a brief summary of the key points raised by reviewers. Rebuttals to individual reviewers can be found at the end of each review.

## Summary

The topic of the paper was deemed “*important and timely*” (Reviewer [Ycp6](https://openreview.net/forum?id=PocqkbIelt&noteId=dkd3AJpstz)), wherein we “***proposed a new learning framework*** *to generate counterfactual explanations (CF) for ML models, which facilitates service providers to specialize CF explanation techniques that can leverage the knowledge of their particular ML model*” (Reviewer [9tKf](https://openreview.net/forum?id=PocqkbIelt&noteId=iDL5HNbhMJs)). As per Reviewer [MUfa](https://openreview.net/forum?id=PocqkbIelt&noteId=CjVOkeXfEA1)’s knowledge, our work “***represents a clearly novel method for the field”***. Reproducibility should be high, and they consider the “***quality of this work to be very high***”. “***Our experiments are solid”***, and our theoretical results add depth to the paper. “***The strengths of the paper far outweigh the weaknesses*” (as per Reviewer [MUfa](https://openreview.net/forum?id=PocqkbIelt&noteId=CjVOkeXfEA1), **who gave us a 10/10 rating**).

There was some confusion regarding the novelty of the paper’s contributions (Reviewer [9tKf](https://openreview.net/forum?id=PocqkbIelt&noteId=iDL5HNbhMJs)), and about perceived ad-hoc design choices (Reviewer [Ycp6](https://openreview.net/forum?id=PocqkbIelt&noteId=dkd3AJpstz)) made in the paper. Reviewer [Mece](https://openreview.net/forum?id=PocqkbIelt&noteId=7t19K9Tr8N) also pointed out a newly published baseline from UAI 2022 that we were not aware of.

In response to these concerns, we have added detailed rebuttals and additional experiments in the Appendix. For example, see our response to Reviewer [Ycp6](https://openreview.net/forum?id=PocqkbIelt&noteId=dkd3AJpstz) about justifications for some of the ad-hoc choices made in the paper. Similarly, see our detailed response to Reviewer [9tKf](https://openreview.net/forum?id=PocqkbIelt&noteId=iDL5HNbhMJs) on clarifying the paper’s novelty. Finally, we implemented the UAI 2022 baseline pointed out by Reviewer [Mece](https://openreview.net/forum?id=PocqkbIelt&noteId=7t19K9Tr8N) and added it as a new baseline in the updated paper version.

---

### Decision · Program_Chairs · 2023-01-20

**Decision:**

Reject

**Justification For Why Not Higher Score:**

Its contributions and novelties are not clear

**Justification For Why Not Lower Score:**

N/A

**Metareview: Summary, Strengths And Weaknesses:**

This paper aims provide counterfactual explanations for neural classification models. Specifically, it proposes to generate counterfactual explanations as a part of the model forward pass by modifying model architecture and the objective. It conducted experiments to compare with some existing methods.

While it is a very important topic, it makes several adhoc choices when setting up the architecture and the loss function. Unfortunately, authors did not explain clearly on the values added and the proposed architecture does not support important criteria like diversity.

**Summary Of Ac-Reviewer Meeting:**

No need to have meeting, as reviewers generally provided negative feedback